# 🦩 Flamingo: a Visual Language Model for Few-Shot Learning

**Jean-Baptiste Alayrac**[*,‡]    **Jeff Donahue**[*]    **Pauline Luc**[*]    **Antoine Miech**[*]

**Iain Barr**[†]    **Yana Hasson**[†]    **Karel Lenc**[†]    **Arthur Mensch**[†]    **Katie Millican**[†]

**Malcolm Reynolds**[†]    **Roman Ring**[†]    **Eliza Rutherford**[†]    **Serkan Cabi**    **Tengda Han**

**Zhitao Gong**    **Sina Samangooei**    **Marianne Monteiro**    **Jacob Menick**

**Sebastian Borgeaud**    **Andrew Brock**    **Aida Nematzadeh**    **Sahand Sharifzadeh**

**Mikolaj Binkowski**    **Ricardo Barreira**    **Oriol Vinyals**    **Andrew Zisserman**

**Karen Simonyan**[*,‡]

[*] **Equal contributions, ordered alphabetically,** [†] **Equal contributions, ordered alphabetically,**
[‡] **Equal senior contributions**

**DeepMind**

## Abstract

Building models that can be rapidly adapted to novel tasks using only a handful of annotated examples is an open challenge for multimodal machine learning research. We introduce Flamingo, a family of Visual Language Models (VLM) with this ability. We propose key architectural innovations to: (i) bridge powerful pretrained vision-only and language-only models, (ii) handle sequences of arbitrarily interleaved visual and textual data, and (iii) seamlessly ingest images or videos as inputs. Thanks to their flexibility, Flamingo models can be trained on large-scale multimodal web corpora containing arbitrarily interleaved text and images, which is key to endow them with in-context few-shot learning capabilities. We perform a thorough evaluation of our models, exploring and measuring their ability to rapidly adapt to a variety of image and video tasks. These include open-ended tasks such as visual question-answering, where the model is prompted with a question which it has to answer; captioning tasks, which evaluate the ability to describe a scene or an event; and close-ended tasks such as multiple-choice visual question-answering. For tasks lying anywhere on this spectrum, a *single* Flamingo model can achieve a new state of the art with few-shot learning, simply by prompting the model with task-specific examples. On numerous benchmarks, *Flamingo* outperforms models fine-tuned on thousands of times more task-specific data.

*Corresponding authors: {jalayrac|jeffdonahue|paulineluc|miech}@deepmind.com*
36th Conference on Neural Information Processing Systems (NeurIPS 2022).

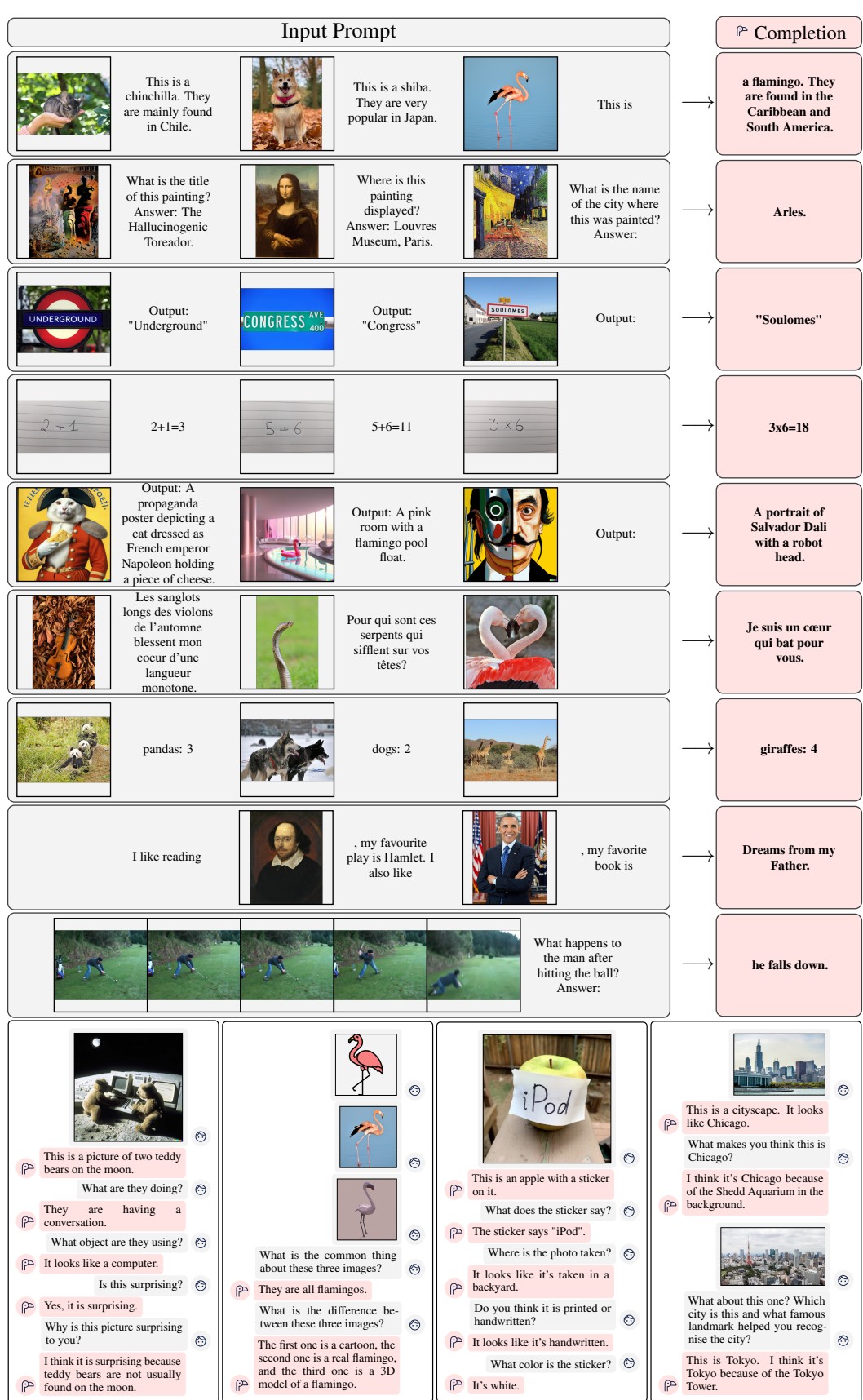

Figure 1: **Selected examples of inputs and outputs obtained from *Flamingo*-80B.** *Flamingo* can rapidly adapt to various image/video understanding tasks with few-shot prompting (top). Out of the box, *Flamingo* is also capable of multi-image visual dialogue (bottom). More examples in Appendix C.

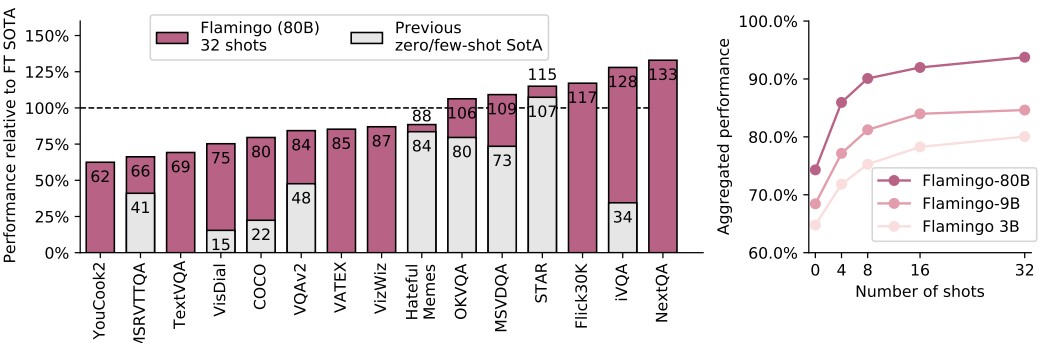

Figure 2: **Flamingo results overview.** *Left*: Our largest model, dubbed *Flamingo*, outperforms state-of-the-art fine-tuned models on 6 of the 16 tasks we consider with no fine-tuning. For the 9 tasks with published few-shot results, *Flamingo* sets the new few-shot state of the art. *Note:* We omit RareAct, our 16th benchmark, as it is a zero-shot benchmark with no available fine-tuned results to compare to. *Right*: Flamingo performance improves with model size and number of shots.

# 1 Introduction

One key aspect of intelligence is the ability to quickly learn to perform a new task given a short instruction [33, 70]. While initial progress has been made towards a similar capability in computer vision, the most widely used paradigm still consists of first pretraining on a large amount of supervised data, before fine-tuning the model on the task of interest [66, 118, 143]. However, successful fine-tuning often requires many thousands of annotated data points. In addition, it often requires careful per-task hyperparameter tuning and is also resource intensive. Recently, multimodal vision-language models trained with a contrastive objective [50, 85] have enabled zero-shot adaptation to novel tasks, without the need for fine-tuning. However, because these models simply provide a similarity score between a text and an image, they can only address limited use cases such as classification, where a finite set of outcomes is provided beforehand. They crucially lack the ability to generate language, which makes them less suitable to more open-ended tasks such as captioning or visual question-answering. Others have explored visually-conditioned language generation [17, 114, 119, 124, 132] but have not yet shown good performance in low-data regimes.

We introduce *Flamingo*, a Visual Language Model (VLM) that sets a new state of the art in few-shot learning on a wide range of open-ended vision and language tasks, simply by being prompted with a few input/output examples, as illustrated in Figure 1. Of the 16 tasks we consider, *Flamingo* also surpasses the fine-tuned state of the art on 6 tasks, despite using orders of magnitude less task-specific training data (see Figure 2). To achieve this, Flamingo takes inspiration from recent work on large language models (LMs) which are good few-shot learners [11, 18, 42, 86]. A single large LM can achieve strong performance on many tasks using only its text interface: a few examples of a task are provided to the model as a prompt, along with a query input, and the model generates a continuation to produce a predicted output for that query. We show that the same can be done for image and video understanding tasks such as classification, captioning, or question-answering: these can be cast as text prediction problems with visual input conditioning. The difference from a LM is that the model must be able to ingest a multimodal prompt containing images and/or videos interleaved with text. Flamingo models have this capability—they are visually-conditioned autoregressive text generation models able to ingest a sequence of text tokens interleaved with images and/or videos, and produce text as output. Flamingo models leverage two complementary pre-trained and frozen models: a vision model which can "perceive" visual scenes and a large LM which performs a basic form of reasoning. Novel architecture components are added in between these models to connect them in a way that preserves the knowledge they have accumulated during computationally intensive pre-training. Flamingo models are also able to ingest high-resolution images or videos thanks to a Perceiver-based [48] architecture that can produce a small fixed number of visual tokens per image/video, given a large and variable number of visual input features.

A crucial aspect for the performance of large LMs is that they are trained on a large amount of text data. This training provides general-purpose generation capabilities that allows these LMs to perform well when prompted with task examples. Similarly, we demonstrate that the way we train the Flamingo models is crucial for their final performance. They are trained on a carefully chosen

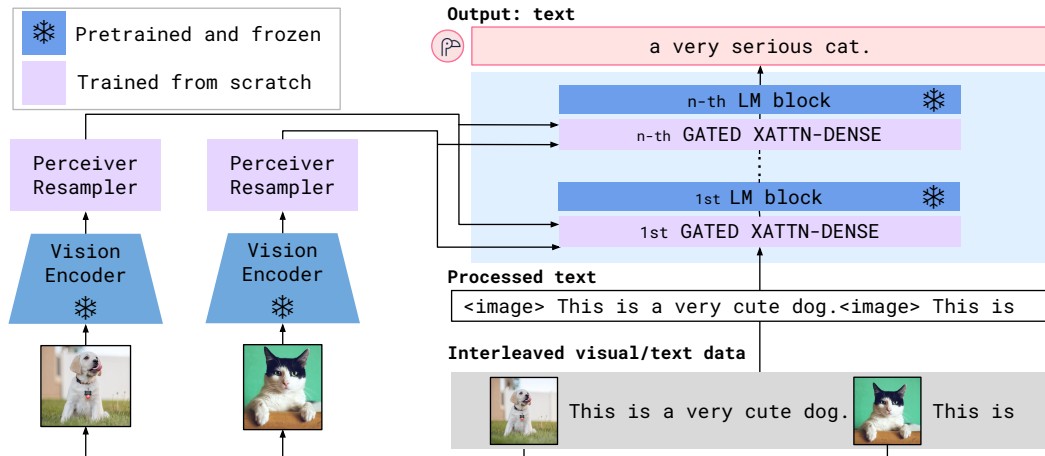

Figure 3: **Flamingo architecture overview.** Flamingo is a family of visual language models (VLMs) that take as input visual data interleaved with text and produce free-form text as output.

mixture of complementary large-scale multimodal data coming only from the web, *without using any data annotated for machine learning purposes*. After this training, a Flamingo model can be directly adapted to vision tasks via simple few-shot learning without any task-specific tuning.

**Contributions.** In summary, our contributions are the following: **(i)** We introduce the Flamingo family of VLMs which can perform various multimodal tasks (such as captioning, visual dialogue, or visual question-answering) from only a few input/output examples. Thanks to architectural innovations, the Flamingo models can efficiently accept arbitrarily interleaved visual data and text as input and generate text in an open-ended manner. **(ii)** We quantitatively evaluate how Flamingo models can be adapted to various tasks via few-shot learning. We notably reserve a large set of held-out benchmarks which have not been used for validation of any design decisions or hyperparameters of the approach. We use these to estimate unbiased few-shot performance. **(iii)** *Flamingo* sets a new state of the art in few-shot learning on a wide array of 16 multimodal language and image/video understanding tasks. On 6 of these 16 tasks, *Flamingo* also outperforms the fine-tuned state of the art despite using only 32 task-specific examples, around 1000 times less task-specific training data than the current state of the art. With a larger annotation budget, *Flamingo* can also be effectively fine-tuned to set a new state of the art on five additional challenging benchmarks: VQAv2, VATEX, VizWiz, MSRVTTQA, and HatefulMemes.

## 2 Approach

This section describes Flamingo: a visual language model that accepts text interleaved with images/videos as input and outputs free-form text. The key architectural components shown in Figure 3 are chosen to leverage pretrained vision and language models and bridge them effectively. First, the Perceiver Resampler (Section 2.1) receives spatio-temporal features from the Vision Encoder (obtained from either an image or a video) and outputs a fixed number of visual tokens. Second, these visual tokens are used to condition the frozen LM using freshly initialised cross-attention layers (Section 2.2) that are interleaved between the pretrained LM layers. These new layers offer an expressive way for the LM to incorporate visual information for the next-token prediction task. Flamingo models the likelihood of text $y$ conditioned on interleaved images and videos $x$ as follows:

$$p(y|x) = \prod_{\ell=1}^{L} p(y_\ell|y_{<\ell}, x_{\leq\ell}), \tag{1}$$

where $y_\ell$ is the $\ell$-th language token of the input text, $y_{<\ell}$ is the set of preceding tokens, $x_{\leq\ell}$ is the set of images/videos preceding token $y_\ell$ in the interleaved sequence and $p$ is parametrized by a Flamingo model. The ability to handle interleaved text and visual sequences (Section 2.3) makes it natural to use Flamingo models for in-context few-shot learning, analogously to GPT-3 with few-shot text prompting. The model is trained on a diverse mixture of datasets as described in Section 2.4.

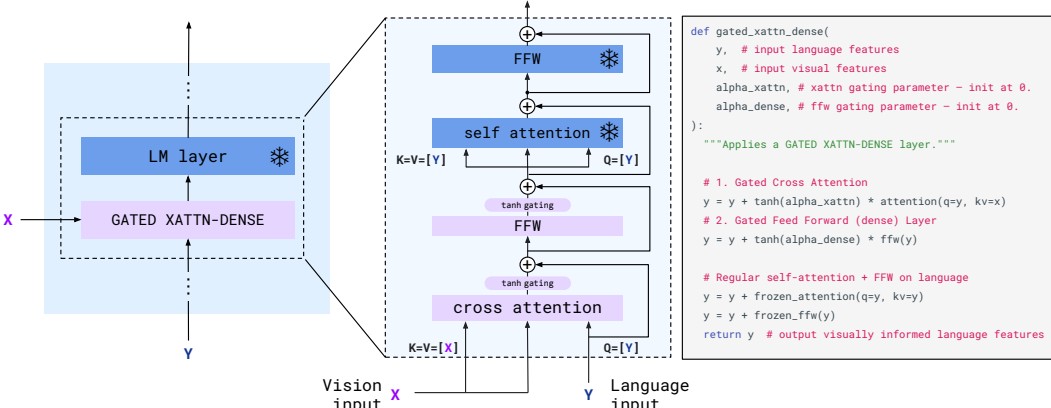

```
def gated_xattn_dense(
    y,  # input language features
    x,  # input visual features
    alpha_xattn, # xattn gating parameter – init at 0.
    alpha_dense, # ffw gating parameter – init at 0.
):
    """Applies a GATED XATTN-DENSE layer."""

    # 1. Gated Cross Attention
    y = y + tanh(alpha_xattn) * attention(q=y, kv=x)
    # 2. Gated Feed Forward (dense) Layer
    y = y + tanh(alpha_dense) * ffw(y)

    # Regular self-attention + FFW on language
    y = y + frozen_attention(q=y, kv=y)
    y = y + frozen_ffw(y)
    return y  # output visually informed language features
```

Figure 4: **GATED XATTN-DENSE layers.** To condition the LM on visual inputs, we insert new cross-attention layers between existing pretrained and frozen LM layers. The keys and values in these layers are obtained from the vision features while the queries are derived from the language inputs. They are followed by dense feed-forward layers. These layers are *gated* so that the LM is kept intact at initialization for improved stability and performance.

## 2.1 Visual processing and the Perceiver Resampler

**Vision Encoder: from pixels to features.** Our vision encoder is a pretrained and frozen Normalizer-Free ResNet (NFNet) [10] – we use the F6 model. We pretrain the vision encoder using a contrastive objective on our datasets of image and text pairs, using the two-term contrastive loss from Radford et al. [85]. We use the output of the final stage, a 2D spatial grid of features that is flattened to a 1D sequence. For video inputs, frames are sampled at 1 FPS and encoded independently to obtain a 3D spatio-temporal grid of features to which learned temporal embeddings are added. Features are then flattened to 1D before being fed to the Perceiver Resampler. More details on the contrastive model training and performance are given in Appendix B.1.3 and Appendix B.3.2, respectively.

**Perceiver Resampler: from varying-size large feature maps to few visual tokens.** This module connects the vision encoder to the frozen language model as shown in Figure 3. It takes as input a variable number of image or video features from the vision encoder and produces a fixed number of visual outputs (64), reducing the computational complexity of the vision-text cross-attention. Similar to Perceiver [48] and DETR [13], we learn a predefined number of latent input queries which are fed to a Transformer and cross-attend to the visual features. We show in our ablation studies (Section 3.3) that using such a vision-language resampler module outperforms a plain Transformer and an MLP. We provide an illustration, more architectural details, and pseudo-code in Appendix A.1.1.

## 2.2 Conditioning frozen language models on visual representations

Text generation is performed by a Transformer decoder, conditioned on the visual representations produced by the Perceiver Resampler. We interleave pretrained and frozen text-only LM blocks with blocks trained from scratch that cross-attend to the visual output from the Perceiver Resampler.

**Interleaving new GATED XATTN-DENSE layers within a frozen pretrained LM.** We freeze the pretrained LM blocks, and insert *gated cross-attention dense* blocks (Figure 4) between the original layers, trained from scratch. To ensure that at initialization, the conditioned model yields the same results as the original language model, we use a $\tanh$-gating mechanism [41]. This multiplies the output of a newly added layer by $\tanh(\alpha)$ before adding it to the input representation from the residual connection, where $\alpha$ is a layer-specific learnable scalar initialized to $0$ [4]. Thus, at initialization, the model output matches that of the pretrained LM, improving training stability and final performance. In our ablation studies (Section 3.3), we compare the proposed GATED XATTN-DENSE layers against recent alternatives [22, 68] and explore the effect of how frequently these additional layers are inserted to trade off between efficiency and expressivity. See Appendix A.1.2 for more details.

**Varying model sizes.** We perform experiments across three models sizes, building on the 1.4B, 7B, and 70B parameter Chinchilla models [42]; calling them respectively *Flamingo*-3B, *Flamingo*-9B and

*Flamingo*-80B. For brevity, we refer to the last as *Flamingo* throughout the paper. While increasing the parameter count of the frozen LM and the trainable vision-text GATED XATTN-DENSE modules, we maintain a fixed-size frozen vision encoder and trainable Perceiver Resampler across the different models (small relative to the full model size). See Appendix B.1.1 for further details.

## 2.3    Multi-visual input support: per-image/video attention masking

The image-causal modelling introduced in Equation (1) is obtained by masking the full text-to-image cross-attention matrix, limiting which visual tokens the model sees at each text token. At a given text token, the model attends to the visual tokens of the image that appeared just before it in the interleaved sequence, rather than to all previous images (formalized and illustrated in Appendix A.1.3). Though the model only *directly* attends to a single image at a time, the dependency on all previous images remains via self-attention in the LM. This single-image cross-attention scheme importantly allows the model to seamlessly generalise to any number of visual inputs, regardless of how many are used during training. In particular, we use only up to 5 images per sequence when training on our interleaved datasets, yet our model is able to benefit from sequences of up to 32 pairs (or "shots") of images/videos and corresponding texts during evaluation. We show in Section 3.3 that this scheme is more effective than allowing the model to cross-attend to all previous images directly.

## 2.4    Training on a mixture of vision and language datasets

We train the Flamingo models on a mixture of three kinds of datasets, all scraped from the web: an interleaved image and text dataset derived from webpages, image-text pairs, and video-text pairs.

**M3W: Interleaved image and text dataset.** The few-shot capabilities of Flamingo models rely on training on interleaved text and image data. For this purpose, we collect the *MultiModal MassiveWeb* (*M3W*) dataset. We extract both text and images from the HTML of approximately 43 million webpages, determining the positions of images relative to the text based on the relative positions of the text and image elements in the Document Object Model (DOM). An example is then constructed by inserting <image> tags in plain text at the locations of the images on the page, and inserting a special <EOC> (*end of chunk*) token (added to the vocabulary and learnt) prior to any image and at the end of the document. From each document, we sample a random subsequence of $L = 256$ tokens and take up to the first $N = 5$ images included in the sampled sequence. Further images are discarded in order to save compute. More details are provided in Appendix A.3.

**Pairs of image/video and text.** For our image and text pairs we first leverage the ALIGN [50] dataset, composed of 1.8 billion images paired with alt-text. To complement this dataset, we collect our own dataset of image and text pairs targeting better quality and longer descriptions: LTIP (Long Text & Image Pairs) which consists of 312 million image and text pairs. We also collect a similar dataset but with videos instead of still images: VTP (Video & Text Pairs) consists of 27 million short videos (approximately 22 seconds on average) paired with sentence descriptions. We align the syntax of paired datasets with the syntax of M3W by prepending <image> and appending <EOC> to each training caption (see Appendix A.3.3 for details).

**Multi-objective training and optimisation strategy.** We train our models by minimizing a weighted sum of per-dataset expected negative log-likelihoods of text, given the visual inputs:

$$\sum_{m=1}^{M} \lambda_m \cdot \mathbb{E}_{(x,y)\sim\mathcal{D}_m} \left[ -\sum_{\ell=1}^{L} \log p(y_\ell | y_{<\ell}, x_{\leq\ell}) \right], \tag{2}$$

where $\mathcal{D}_m$ and $\lambda_m$ are the $m$-th dataset and its weighting, respectively. Tuning the per-dataset weights $\lambda_m$ is key to performance. We accumulate gradients over all datasets, which we found outperforms a "round-robin" approach [17]. We provide further training details and ablations in Appendix B.1.2.

## 2.5    Task adaptation with few-shot in-context learning

Once Flamingo is trained, we use it to tackle a visual task by conditioning it on a multimodal interleaved prompt. We evaluate the ability of our models to rapidly adapt to new tasks using **in-context learning**, analogously to GPT-3 [11], by interleaving support example pairs in the form of $(image, text)$ or $(video, text)$, followed by the query visual input, to build a prompt (details in

| Method | FT | Shot | OKVQA (I) | VQAv2 (I) | COCO (I) | MSVDQA (V) | VATEX (V) | VizWiz (I) | Flick30K (I) | MSRVTTQA (V) | iVQA (V) | YouCook2 (V) | STAR (V) | VisDial (I) | TextVQA (I) | NextQA (I) | HatefulMemes (I) | RareAct (V) |
|---|---|---|---|---|---|---|---|---|---|---|---|---|---|---|---|---|---|---|
| Zero/Few shot SOTA | ✗ | (X) | [34] 43.3 (16) | [114] 38.2 (4) | [124] 32.2 (0) | [58] 35.2 (0) | - | - | - | [58] 19.2 (0) | [135] 12.2 (0) | - | [143] 39.4 (0) | [79] 11.6 (0) | - | - | [85] 66.1 (0) | [85] 40.7 (0) |
| *Flamingo*-3B | ✗ | 0 | 41.2 | 49.2 | 73.0 | 27.5 | 40.1 | 28.9 | 60.6 | 11.0 | 32.7 | 55.8 | 39.6 | 46.1 | 30.1 | 21.3 | 53.7 | 58.4 |
|  | ✗ | 4 | 43.3 | 53.2 | 85.0 | 33.0 | 50.0 | 34.0 | 72.0 | 14.9 | 35.7 | 64.6 | 41.3 | 47.3 | 32.7 | 22.4 | 53.6 | - |
|  | ✗ | 32 | 45.9 | 57.1 | 99.0 | 42.6 | 59.2 | 45.5 | 71.2 | 25.6 | 37.7 | 76.7 | 41.6 | 47.3 | 30.6 | 26.1 | 56.3 | - |
| *Flamingo*-9B | ✗ | 0 | 44.7 | 51.8 | 79.4 | 30.2 | 39.5 | 28.8 | 61.5 | 13.7 | 35.2 | 55.0 | 41.8 | 48.0 | 31.8 | 23.0 | 57.0 | 57.9 |
|  | ✗ | 4 | 49.3 | 56.3 | 93.1 | 36.2 | 51.7 | 34.9 | 72.6 | 18.2 | 37.7 | 70.8 | **42.8** | 50.4 | 33.6 | 24.7 | 62.7 | - |
|  | ✗ | 32 | 51.0 | 60.4 | 106.3 | 47.2 | 57.4 | 44.0 | 72.8 | 29.4 | 40.7 | 77.3 | 41.2 | 50.4 | 32.6 | 28.4 | 63.5 | - |
| *Flamingo* | ✗ | 0 | 50.6 | 56.3 | 84.3 | 35.6 | 46.7 | 31.6 | 67.2 | 17.4 | 40.7 | 60.1 | 39.7 | 52.0 | 35.0 | 26.7 | 46.4 | **60.8** |
|  | ✗ | 4 | 57.4 | 63.1 | 103.2 | 41.7 | 56.0 | 39.6 | 75.1 | 23.9 | 44.1 | 74.5 | 42.4 | **55.6** | 36.5 | 30.8 | 68.6 | - |
|  | ✗ | 32 | **57.8** | **67.6** | **113.8** | **52.3** | **65.1** | **49.8** | **75.4** | **31.0** | **45.3** | **86.8** | 42.2 | **55.6** | **37.9** | **33.5** | **70.0** | - |
| Pretrained FT SOTA | ✔ | (X) | 54.4 [34] (10K) | 80.2 [140] (444K) | 143.3 [124] (500K) | 47.9 [28] (27K) | 76.3 [153] (500K) | 57.2 [65] (20K) | 67.4 [150] (30K) | 46.8 [51] (130K) | 35.4 [135] (6K) | 138.7 [132] (10K) | 36.7 [128] (46K) | 75.2 [79] (123K) | 54.7 [137] (20K) | 25.2 [129] (38K) | 79.1 [62] (9K) | - |

Table 1: **Comparison to the state of the art.** A *single* Flamingo model reaches the state of the art on a wide array of image (**I**) and video (**V**) understanding tasks with few-shot learning, significantly outperforming previous best zero- and few-shot methods with as few as four examples. More importantly, using only 32 examples and without adapting any model weights, Flamingo *outperforms* the current best methods – fine-tuned on thousands of annotated examples – on seven tasks. Best few-shot numbers are in **bold**, best numbers overall are underlined.

Appendix A.2). We perform **open-ended** evaluations using beam search for decoding, and **close-ended** evaluations using our model's log-likelihood to score each possible answer. We explore **zero-shot generalization** by prompting the model with two text-only examples from the task, with no corresponding images. Evaluation hyperparameters and additional details are given in Appendix B.1.5.

# 3   Experiments

Our goal is to develop models that can rapidly adapt to diverse and challenging tasks. For this, we consider a wide array of 16 popular multimodal image/video and language benchmarks. In order to validate model design decisions during the course of the project, 5 of these benchmarks were used as part of our development (DEV) set: COCO, OKVQA, VQAv2, MSVDQA and VATEX. Performance estimates on the DEV benchmarks may be biased, as a result of model selection. We note that this is also the case for prior work which makes use of similar benchmarks to validate and ablate design decisions. To account for this, we report performance on an additional set of 11 benchmarks, spanning captioning, video question-answering, as well as some less commonly explored capabilities such as visual dialogue and multi-choice question-answering tasks. The evaluation benchmarks are described in Appendix B.1.4. We keep all evaluation hyperparameters fixed across all benchmarks. Depending on the task, we use four few-shot prompt templates we describe in more detail in Appendix B.1.5. We emphasize that *we do not validate any design decisions on these 11 benchmarks* and use them solely to estimate unbiased few-shot learning performance of our models.

Concretely, estimating few-shot learning performance of a model involves prompting it with a set of *support* samples and evaluating it on a set of *query* samples. For the DEV benchmarks that are used both to validate design decisions and hyperparameters, as well as to report final performance, we therefore use four subsets: *validation support*, *validation query*, *test support* and *test query*. For other benchmarks, we need only the latter two. We report in Appendix B.1.4 how we form these subsets.

We report the results of the Flamingo models on few-shot learning in Section 3.1. Section 3.2 gives *Flamingo* fine-tuned results. An ablation study is given in Section 3.3. Appendix B.2 provides more results including Flamingo's performance on the ImageNet and Kinetics700 classification tasks, and on our contrastive model's performance. Appendix C includes additional qualitative results.

## 3.1   Few-shot learning on vision-language tasks

**Few-shot results.** Results are given in Table 1. *Flamingo* outperforms by a large margin *all* previous zero-shot or few-shot methods on the 16 benchmarks considered. This is achieved with as few as four examples per task, demonstrating practical and efficient adaptation of vision models to new tasks. More importantly, *Flamingo* is often competitive with state-of-the-art methods additionally fine-tuned

| Method | VQAV2 | | COCO | VATEX | VizWiz | | MSRVTTQA | VisDial | | YouCook2 | TextVQA | | HatefulMemes |
|---|---|---|---|---|---|---|---|---|---|---|---|---|---|
| | test-dev | test-std | test | test | test-dev | test-std | test | valid | test-std | valid | valid | test-std | test seen |
| ⚘ 32 shots | 67.6 | - | 113.8 | 65.1 | 49.8 | - | 31.0 | 56.8 | - | 86.8 | 36.0 | - | 70.0 |
| ⚘ Fine-tuned | **82.0** | **82.1** | 138.1 | **84.2** | **65.7** | **65.4** | 47.4 | 61.8 | 59.7 | 118.6 | **57.1** | 54.1 | **86.6** |
| SotA | 81.3† | 81.3† | **149.6†** | 81.4† | 57.2† | 60.6† | 46.8 | 75.2 | 75.4† | 138.7 | 54.7 | 73.7 | 84.6† |
| | [133] | [133] | [119] | [153] | [65] | [65] | [51] | [79] | [123] | [132] | [137] | [84] | [152] |

Table 2: **Comparison to SotA when fine-tuning *Flamingo*.** We fine-tune *Flamingo* on all nine tasks where *Flamingo* does not achieve SotA with few-shot learning. *Flamingo* sets a new SotA on five of them, outperforming methods (marked with †) that use tricks such as model ensembling or domain-specific metric optimisation (e.g., CIDEr optimisation).

| | Ablated setting | *Flamingo*-3B original value | Changed value | Param. count ↓ | Step time ↓ | COCO CIDEr↑ | OKVQA top1↑ | VQAv2 top1↑ | MSVDQA top1↑ | VATEX CIDEr↑ | Overall score↑ |
|---|---|---|---|---|---|---|---|---|---|---|---|
| | | *Flamingo*-3B model | | 3.2B | 1.74s | 86.5 | 42.1 | 55.8 | 36.3 | 53.4 | 70.7 |
| **(i)** | Training data | All data | w/o Video-Text pairs | 3.2B | 1.42s | 84.2 | 43.0 | 53.9 | 34.5 | 46.0 | 67.3 |
| | | | w/o Image-Text pairs | 3.2B | 0.95s | 66.3 | 39.2 | 51.6 | 32.0 | 41.6 | 60.9 |
| | | | Image-Text pairs→ LAION | 3.2B | 1.74s | 79.5 | 41.4 | 53.5 | 33.9 | 47.6 | 66.4 |
| | | | w/o M3W | 3.2B | 1.02s | 54.1 | 36.5 | 52.7 | 31.4 | 23.5 | 53.4 |
| **(ii)** | Optimisation | Accumulation | Round Robin | 3.2B | 1.68s | 76.1 | 39.8 | 52.1 | 33.2 | 40.8 | 62.9 |
| **(iii)** | Tanh gating | ✓ | ✗ | 3.2B | 1.74s | 78.4 | 40.5 | 52.9 | 35.9 | 47.5 | 66.5 |
| **(iv)** | Cross-attention architecture | GATED XATTN-DENSE | VANILLA XATTN | 2.4B | 1.16s | 80.6 | 41.5 | 53.4 | 32.9 | 50.7 | 66.9 |
| | | | GRAFTING | 3.3B | 1.74s | 79.2 | 36.1 | 50.8 | 32.2 | 47.8 | 63.1 |
| **(v)** | Cross-attention frequency | Every | Single in middle | 2.0B | 0.87s | 71.5 | 38.1 | 50.2 | 29.1 | 42.3 | 59.8 |
| | | | Every 4th | 2.3B | 1.02s | 82.3 | 42.7 | 55.1 | 34.6 | 50.8 | 68.8 |
| | | | Every 2nd | 2.6B | 1.24s | 83.7 | 41.0 | 55.8 | 34.5 | 49.7 | 68.2 |
| **(vi)** | Resampler | Perceiver | MLP | 3.2B | 1.85s | 78.6 | 42.2 | 54.7 | 35.2 | 44.7 | 66.6 |
| | | | Transformer | 3.2B | 1.81s | 83.2 | 41.7 | 55.6 | 31.5 | 48.3 | 66.7 |
| **(vii)** | Vision encoder | NFNet-F6 | CLIP ViT-L/14 | 3.1B | 1.58s | 76.5 | 41.6 | 53.4 | 33.2 | 44.5 | 64.9 |
| | | | NFNet-F0 | 2.9B | 1.45s | 73.8 | 40.5 | 52.8 | 31.1 | 42.9 | 62.7 |
| **(viii)** | Freezing LM | ✓ | ✗ (random init) | 3.2B | 2.42s | 74.8 | 31.5 | 45.6 | 26.9 | 50.1 | 57.8 |
| | | | ✗ (pretrained) | 3.2B | 2.42s | 81.2 | 33.7 | 47.4 | 31.0 | 53.9 | 62.7 |

Table 3: **Ablation studies.** Each row should be compared to the baseline Flamingo run (top row). Step time measures the time spent to perform gradient updates on all training datasets.

on up to hundreds of thousands of annotated examples. On six tasks, *Flamingo* even outperforms the fine-tuned SotA despite using a *single* set of model weights and only 32 task-specific examples. Finally, despite having only used the DEV benchmarks for design decisions, our results generalize well to the other benchmarks, confirming the generality of our approach.

**Scaling with respect to parameters and shots.** As shown in Figure 2, the larger the model, the better the few-shot performance, similar to GPT-3 [11]. The performance also improves with the number of shots. We further find that the largest model better exploits larger numbers of shots. Interestingly, even though our Flamingo models were trained with sequences limited to only 5 images on *M3W*, they are still able to benefit from up to 32 images or videos during inference. This demonstrates the flexibility of the Flamingo architecture for processing a variable number of videos or images.

### 3.2 Fine-tuning *Flamingo* as a pretrained vision-language model

While not the main focus of our work, we verify that when given more data, Flamingo models can be adapted to a task by fine-tuning their weights. In Table 2, we explore fine-tuning our largest model, *Flamingo*, for a given task with no limit on the annotation budget. In short, we do so by fine-tuning the model on a short schedule with a small learning rate by additionally unfreezing the vision backbone to accommodate a higher input resolution (details in Appendix B.2.2). We find that we can improve results over our previously presented in-context few-shot learning results, setting a new state of the art on five additional tasks: VQAv2, VATEX, VizWiz, MSRVTTQA, and HatefulMemes.

### 3.3 Ablation studies

In Table 3, we report our ablation results using *Flamingo*-3B on the *validation* subsets of the five DEV benchmarks with 4 shots. Note that we use smaller batch sizes and a shorter training schedule compared to the final models. The **Overall score** is obtained by dividing each benchmark score by its state-of-the-art (SotA) performance from Table 1 and averaging the results. More details and results are given in Appendix B.3 and Table 10.

**Importance of the training data mixture.** As shown in row **(i)**, getting the right training data plays a crucial role. In fact, removing the interleaved image-text dataset *M3W* leads to a *decrease of more*

*than* 17% in performance while removing the conventional paired image-text pairs also decreases performance (by 9.8%), demonstrating the need for different types of datasets. Moreover, removing our paired video-text dataset negatively affects performance on all video tasks. We ablate replacing our image-text pairs (ITP) by the publicly available LAION-400M dataset [96], which leads to a slight degradation in performance. We show in row **(ii)** the importance of our gradient accumulation strategy compared to using round-robin updates [17].

**Visual conditioning of the frozen LM.** We ablate the use of the 0-initialized tanh gating when merging the cross-attention output to the frozen LM output in row **(iii)**. Without it, we see a drop of 4.2% in our overall score. Moreover, we have noticed that disabling the 0-initialized tanh gating leads to training instabilities. Next, we ablate different conditioning architectures in row **(iv)**. VANILLA XATTN, refers to the vanilla cross-attention from the original Transformer decoder [115]. In the GRAFTING approach from [68], the frozen LM is used as is with no additional layers inserted, and a stack of interleaved self-attention and cross-attention layers that take the frozen LM output are learnt from scratch. Overall, we show that our GATED XATTN-DENSE conditioning approach works best.

**Compute/Memory vs. performance trade-offs.** In row **(v)**, we ablate the frequency at which we add new GATED XATTN-DENSE blocks. Although adding them at every layer is better, it significantly increases the number of trainable parameters and time complexity of the model. Notably, inserting them every fourth block accelerates training by 66% while only decreasing the overall score by 1.9%. In light of this trade-off, we maximize the number of added layers under hardware constraints and add a GATED XATTN-DENSE every fourth layer for *Flamingo*-9B and every seventh for *Flamingo*-80B. We further compare in row **(vi)** the Perceiver Resampler to a MLP and a vanilla Transformer given a parameter budget. Both underperform the Perceiver Resampler while also being slower.

**Vision encoder.** In row **(vii)**, we compare our NFNet-F6 vision encoder pretrained with contrastive learning (details in Appendix B.1.3) to the publicly available CLIP ViT-L/14 [85] model trained at 224 resolution. Our NFNet-F6 has a +5.8% advantage over the CLIP ViT-L/14 and +8.0% over a smaller NFNet-F0 encoder, which highlights the importance of using a strong vision backbone.

**Freezing LM components prevents catastrophic forgetting.** We verify the importance of freezing the LM layers at training in row **(viii)**. If trained from scratch, we observe a large performance decrease of −12.9%. Interestingly, fine-tuning our pretrained LM also leads to a drop in performance of −8.0%. This indicates an instance of "catastrophic forgetting" [71], in which the model progressively forgets its pretraining while training on a new objective. In our setting, freezing the language model is a better alternative to training with the pre-training dataset (MassiveText) in the mixture.

## 4 Related work

**Language modelling and few-shot adaptation.** Language modelling has recently made substantial progress following the introduction of Transformers [115]. The paradigm of first pretraining on a vast amount of data followed by an adaptation on a downstream task has become standard [11, 23, 32, 44, 52, 75, 87, 108]. In this work, we build on the 70B Chinchilla language model [42] as the base LM for *Flamingo*. Numerous works have explored techniques to adapt language models to novel tasks using a few examples. These include adding small adapter modules [43], fine-tuning a small part of the LM [141], showing in-context examples in the prompt [11], or optimizing the prompt [56, 60] through gradient descent. In this paper, we take inspiration from the in-context [11] few-shot learning technique instead of more involved few-shot learning approaches based on metric learning [24, 103, 112, 117] or meta-learning [6, 7, 27, 31, 91, 155].

**When language meets vision.** These LM breakthroughs have been influential for vision-language modelling. In particular, BERT [23] inspired a large body of vision-language work [16, 28, 29, 38, 59, 61, 66, 101, 106, 107, 109, 118, 121, 142, 143, 151]. We differ from these approaches as Flamingo models do not require fine-tuning on new tasks. Another family of vision-language models is based on contrastive learning [2, 5, 49, 50, 57, 74, 82, 85, 138, 140, 146]. Flamingo differs from contrastive models as it can generate text, although we build and rely upon them for our vision encoder. Similar to our work are VLMs able to generate text in an autoregressive manner [19, 25, 45, 67, 116]. Concurrent works [17, 58, 119, 124, 154] also propose to formulate numerous vision tasks as text generation problems. Building on top of powerful pretrained language models has been explored in several recent works. One recent line of work [26, 68, 78, 114, 136, 144] proposes to freeze the pretrained LM weights to prevent catastrophic forgetting [71]. We follow this idea by freezing the

Chinchilla LM layers [42] and adding learnable layers within the frozen LM. We differ from prior work by introducing the first LM that can ingest arbitrarily interleaved images, videos, and text.

**Web-scale vision and language training datasets.** Manually annotated vision and language datasets are costly to obtain and thus relatively small (10k-100k) in scale [3, 15, 69, 122, 129, 139]. To alleviate this lack of data, numerous works [14, 50, 98, 110] automatically scrape readily available paired vision-text data. In addition to such paired data, we show the importance of also training on entire multimodal webpages containing interleaved images and text as a single sequence. Concurrent work CM3 [1] proposes to generate HTML markup from pages, while we simplify the text prediction task by only generating plain text. We emphasize few-shot learning and vision tasks while CM3 [1] primarily evaluates on language-only benchmarks in a zero-shot or fine-tuned setup.

# 5    Discussion

**Limitations.** First, our models build on pretrained LMs, and as a side effect, directly inherit their weaknesses. For example, LM priors are generally helpful, but may play a role in occasional hallucinations and ungrounded guesses. Furthermore, LMs generalise poorly to sequences longer than the training ones. They also suffer from poor sample efficiency during training. Addressing these issues can accelerate progress in the field and enhance the abilities of VLMs like Flamingo.

Second, the classification performance of Flamingo lags behind that of state-of-the-art contrastive models [82, 85]. These models directly optimize for text-image retrieval, of which classification is a special case. In contrast, our models handle a wider range of tasks, such as open-ended ones. A unified approach to achieve the best of both worlds is an important research direction.

Third, in-context learning has significant advantages over gradient-based few-shot learning methods, but also suffers from drawbacks depending on the characteristics of the application at hand. We demonstrate the effectiveness of in-context learning when access is limited to only a few dozen examples. In-context learning also enables simple deployment, requiring only inference, generally with no hyperparameter tuning needed. However, in-context learning is known to be highly sensitive to various aspects of the demonstrations [80, 148], and its inference compute cost and absolute performance scale poorly with the number of shots beyond this low-data regime. There may be opportunities to combine few-shot learning methods to leverage their complementary benefits. We discuss the limitations of our work in more depth in Appendix D.1.

**Societal impacts.** In terms of societal impacts, *Flamingo* offers a number of benefits while carrying some risks. Its ability to rapidly adapt to a broad range of tasks have the potential to enable non-expert users to obtain good performance in data-starved regimes, lowering the barriers to both beneficial and malicious applications. *Flamingo* is exposed to the same risks as large language models, such as outputting offensive language, propagating social biases and stereotypes, as well as leaking private information [42, 126]. Its ability to additionally handle visual inputs poses specific risks such as gender and racial biases relating to the contents of the input images, similar to a number of visual recognition systems [12, 21, 37, 97, 147]. We refer the reader to Appendix D.2 for a more extensive discussion of the societal impacts of our work, both positive and negative; as well as mitigation strategies and early investigations of risks relating to racial or gender bias and toxic outputs. Finally we note that, following prior work focusing on language models [72, 81, 111], the few-shot capabilities of Flamingo could be useful for mitigating such risks.

**Conclusion.** We proposed Flamingo, a general-purpose family of models that can be applied to image and video tasks with minimal task-specific training data. We also qualitatively explored interactive abilities of *Flamingo* such as "chatting" with the model, demonstrating flexibility beyond traditional vision benchmarks. Our results suggest that connecting pre-trained large language models with powerful visual models is an important step towards general-purpose visual understanding.

**Acknowledgments and Disclosure of Funding.**    This research was funded by DeepMind. We would like to thank many colleagues for useful discussions, suggestions, feedback, and advice, including: Samuel Albanie, Relja Arandjelović, Kareem Ayoub, Lorrayne Bennett, Adria Recasens Continente, Tom Eccles, Nando de Freitas, Sander Dieleman, Conor Durkan, Aleksa Gordić, Raia Hadsell, Will Hawkins, Lisa Anne Hendricks, Felix Hill, Jordan Hoffmann, Geoffrey Irving, Drew Jaegle, Koray Kavukcuoglu, Agustin Dal Lago, Mateusz Malinowski, Soňa Mokrá, Gaby Pearl, Toby Pohlen, Jack Rae, Laurent Sifre, Francis Song, Maria Tsimpoukelli, Gregory Wayne, and Boxi Wu.

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
