# Appendix

We provide an overview of the Appendix below.

**Method (Appendix A).** We first provide additional details about our model in Appendix A.1:

- An illustration and pseudo-code for the Perceiver Resampler (described in Section 2.1) is provided in Appendix A.1.1 and Figure 5.
- A similar illustration is provided for the GATED XATTN-DENSE layer of Section 2.2 in Appendix A.1.2 and Figure 4.
- Details on our implementation of the multi-image/video attention mechanism (Section 2.3) are given in Appendix A.1.3.
- Hyperparameters for all model architectures are given in Appendix A.1.4.

We then explain how we evaluate our models using in-context few-shot learning in Appendix A.2. This includes details on how we build the few-shot prompt, how we get predictions for open- and close-ended tasks, how we obtain the zero-shot numbers, and how we leverage retrieval and ensembling to take advantage of more annotated examples.

Finally, in Appendix A.3, we provide more details on our training datasets:

- Collection of M3W in Appendix A.3.1,
- How we process M3W samples during training in Appendix A.3.2,
- Collection of LTIP and VTP in Appendix A.3.3,
- Deduplication strategy we employ to ensure that there is no leakage between our training and evaluation datasets in Appendix A.3.4.

**Experiments (Appendix B).** We first provide additional training and evaluation details in Appendix B.1, including:

- Details on *Flamingo*-3B, *Flamingo*-9B and *Flamingo* in Appendix B.1.1,
- The training hyperparameters in Appendix B.1.2,
- More details on the Contrastive model pretraining in Appendix B.1.3,
- Details on our evaluation benchmarks and splits in Appendix B.1.4,
- A discussion on the few-shot learning hyperparameters in Appendix B.1.5,
- The dialogue prompt used in the qualitative dialogue examples shown in Figure 1 and Figure 11 in Appendix B.1.6.

Next, we give additional results obtained by our models in Appendix B.2 including the performance of the Flamingo models on classification tasks in Appendix B.2.1, detailed fine-tuning results in Appendix B.2.2, and zero-shot results from our contrastive models (Appendix B.2.3).

Finally, we provide more ablation studies in Appendix B.3 for both the Flamingo models (Appendix B.3.1) and our contrastive pretrained Visual Encoders (Appendix B.3.2).

**Qualitative results (Appendix C).** More qualitative results are given in Appendix C: Figure 10 (single image sample), Figure 11 (dialogue examples), and Figure 12 (video examples).

**Discussion (Appendix D).** We provide a more complete discussion on our work, including limitations, failure cases, broader impacts and societal impacts of our work in Appendix D.

**Model card (Appendix E).** The *Flamingo* model card is provided in Appendix E.

**Datasheets (Appendix F).** Datasheets for M3W, LTIP and VTP are respectively given in Appendix F.1, Appendix F.2.1 and Appendix F.2.2.

**Credit for visual content (Appendix G).** We provide attribution for all visual illustrations used in the paper in Appendix G.

[Figure]

```
def perceiver_resampler(
    x_f,  # The [T, S, d] visual features (T=time, S=space)
    time_embeddings,  # The [T, 1, d] time pos embeddings.
    x,  # R learned latents of shape [R, d]
    num_layers,  # Number of layers
):
    """The Perceiver Resampler model."""

    # Add the time position embeddings and flatten.
    x_f = x_f + time_embeddings
    x_f = flatten(x_f)  # [T, S, d] -> [T * S, d]
    # Apply the Perceiver Resampler layers.
    for i in range(num_layers):
        # Attention.
        x = x + attention_i(q=x, kv=concat([x_f, x]))
        # Feed forward.
        x = x + ffw_i(x)
    return x
```

Figure 5: **The Perceiver Resampler** module maps a *variable* size grid of spatio-temporal visual features output by the Vision Encoder to a *fixed* number of output tokens (five in the figure), independently from the input image resolution or the number of input video frames. This transformer has a set of learned latent vectors as queries, and the keys and values are a concatenation of the spatio-temporal visual features with the learned latent vectors.

## A    Method

### A.1    Model details

#### A.1.1    Perceiver Resampler

Expanding on our brief description in Section 2.1, Figure 5 provides an illustration of our Perceiver Resampler processing an example video, together with pseudo-code. Our Perceiver Resampler is similar in spirit to the Perceiver models proposed by Jaegle et al. [48]. We learn a predefined number of latent input queries, and cross-attend to the flattened visual features $X_f$. These visual features $X_f$ are obtained by first adding a learnt temporal position encoding to each feature within a given video frame (an image being considered as a single-frame video). Note that we only use temporal encodings and no explicit spatial grid position encodings; we did not observe improvements from the latter. This rationale behind is likely that CNNs, such as our NFNet encoder, are known to implicitly include spatial information channel-wise [47]. The visual features are then flattened and concatenated as illustrated in Figure 5. The number of output tokens of the Perceiver Resampler is equal to the number of learnt latent queries. Unlike in DETR and Perceiver, the keys and values computed from the learnt latents are concatenated to the keys and values obtained from $X_f$, which we found to perform slightly better.

#### A.1.2    GATED XATTN-DENSE details

We provide in Figure 4 an illustration of a GATED XATTN-DENSE block and how it connects to a frozen LM block, together with pseudo-code.

We also plot in Figure 6 the evolution of the absolute value of the $\tanh$ gating values as a function of training progress (from $0\%$ to $100\%$) at different layers of the LM stack for the *Flamingo*-3B model composed of 24 LM layers. All layers of the frozen LM stack seem to utilize the visual information as the $\tanh$ gating absolute values quickly grow in absolute value from their 0 initializations. We also note that the absolute values seem to grow with the depth. However, it is difficult to draw strong conclusions from this observation: the scale of the activations before gating may also vary with depth.

[Figure]

(a) Attention tanh gating                      (b) FFW tanh gating.

Figure 6: Evolution of the absolute value of the tanh gating at different layers of *Flamingo*-3B.

[Figure]

Figure 7: **Interleaved visual data and text support.** Given text interleaved with images/videos, e.g. coming from a webpage, we first process the text by inserting `<image>` tags at the locations of the visual data in the text as well as special tokens (`<BOS>` for "beginning of sequence" or `<EOC>` for "end of chunk"). Images are processed independently by the Vision Encoder and Perceiver Resampler to extract visual tokens. At a given text token, the model only cross-attends to the visual tokens corresponding to the last preceding image/video. $\phi$ indicates which image/video a text token can attend or 0 when no image/video is preceding. In practice, this selective cross-attention is achieved through masking – illustrated here with the dark blue entries (unmasked/visible) and light blue entries (masked).

Future work is required to better understand the effect of these added layers on the optimization dynamics and on the model itself.

### A.1.3   Multi-visual input support

We illustrate in Figure 7 the masking approach we use to limit the number of visual tokens that a certain text token sees. We also formalize our notation for the interleaved sequences of images/videos and text.

**Interleaved sequences of visual data and text.** We consider interleaved image/video and text examples: each example holds a sequence of text $y$, a sequence of images/videos $x$, and the sequence of positions of the images in the text. Based on the visual data positions, we define a function $\phi : [1, L] \mapsto [0, N]$ that assigns to each text position the index of the last image/video appearing before this position (or 0 if no visual data appears before the position). The function $\phi$ defines which visual inputs we consider usable to predict token $\ell$ in Equation (1): the set of preceding tokens $y_{<\ell} \triangleq (y_1, \ldots, y_{\ell-1})$, and the set of preceding images/videos $x_{\leq\ell} \triangleq \{x_i | i \leq \phi(\ell)\}$.

### A.1.4   Transformer architecture

We list in Table 4 the number of layers ($L$), the hidden dimension ($D$), the number of heads ($H$), and the FFW activation (Act.) used for each transformer component of our Flamingo models. The dimension of keys and values in each configuration is given by $D/H$ (96 for the Perceiver Resampler; 128 for GATED XATTN-DENSE and the frozen LM), and the hidden dimension of each feed-forward

| | Perceiver Resampler | | | | GATED XATTN-DENSE | | | | Frozen LM | | | |
|---|---|---|---|---|---|---|---|---|---|---|---|---|---|
| | L | D | H | Act. | L | D | H | Act. | L | D | H | Act. |
| *Flamingo*-3B | 6 | 1536 | 16 | Sq. ReLU | 24 | 2048 | 16 | Sq. ReLU | 24 | 2048 | 16 | GeLU |
| *Flamingo*-9B | 6 | 1536 | 16 | Sq. ReLU | 10 | 4096 | 32 | Sq. ReLU | 40 | 4096 | 32 | GeLU |
| *Flamingo* | 6 | 1536 | 16 | Sq. ReLU | 12 | 8192 | 64 | Sq. ReLU | 80 | 8192 | 64 | GeLU |

Table 4: Hyper-parameters for the Flamingo models' transformers. The hidden size of each feed-forward MLP is $4D$. **L**: number of layers, **D**: transformer hidden size, **H**: number of heads, **Act.**: FFW activation, **Sq. ReLU**: Squared ReLU [104].

[Figure]

Figure 8: **Few-shot interleaved prompt generation.** Given some task-specific few-shot examples (a.k.a. support examples) and a query for which Flamingo should make a prediction, we build the prompt by interleaving images with their corresponding texts. We introduce some formatting to do this, prepending "Output:" to the expected response for all vision-to-text tasks or prompting in the format "Question: {question} Answer: {answer}" for visual question-answering tasks.

MLP is $4D$. Note that the frozen LM was trained with the GeLU activation [39], while the remaining trainable transformer layers use the Squared ReLU activation [104], which we found to outperform GeLU.

## A.2 In-context few-shot evaluation details

**In-context learning with Flamingo models.** We evaluate the ability of our models to rapidly adapt to new tasks using in-context learning, following an analogous approach to the one used in GPT-3 [11]. In detail, we are given a set of support examples in the form of $(image, text)$ or $(video, text)$ (where the $image$ or $video$ is the input visual and the $text$ is the expected response and any additional task-specific information, e.g., a question) and a single visual query for which we want our model to make a prediction. Given this, we build a multimodal prompt by concatenating the support examples followed by the visual query as illustrated by Figure 8. Unless specified otherwise, we choose the concatenation order at random.

**Open-ended and close-ended evaluations.** In an open-ended setting, the model's sampled text following the query image is then taken as its prediction for the image, stopping at the first <EOC> ("end of chunk") token prediction. Unless specified otherwise, we always use beam search with a beam size of 3. In a close-ended setting, all possible outputs are independently appended to the query image, and we score each of the resulting sequences using the log-likelihood estimated by our model. These scores are then used to rank the candidate outputs in decreasing order, from most confident to least confident.

[Figure] [Figure] [Figure] [Figure]

Image-Text Pairs dataset
[N=1, T=1, H, W, C]

Video-Text Pairs dataset
[N=1, T>1, H, W, C]

Multi-Modal Massive Web (M3W) dataset
[N>1, T=1, H, W, C]

Figure 9: **Training datasets.** Mixture of training datasets of different formats. $N$ corresponds to the number of visual inputs for a single example. For paired image (or video) and text datasets, $N = 1$. $T$ is the number of video frames ($T = 1$ for images). $H$, $W$, and $C$ are height, width and color channels.

**Zero-shot generalization.** In the absence of few-shot examples, approaches commonly rely on prompt engineering [85] to condition the model at inference using a suitable natural language description of the task. Validation of such prompts can significantly impact performance but requires access to a number of annotated examples and cannot therefore be considered truly zero-shot. Furthermore, Perez et al. [80] have shown that such validation procedures are generally not robust with access to only a handful of samples during validation. To report zero-shot performance in our work, we instead build a prompt with *two examples* from the downstream tasks *where we remove their corresponding images or videos*. For example, for the task illustrated at the top of Figure 8, the prompt would be "`<BOS>Output: This is a cat wearing sunglasses.<EOC>Output: Three elephants walking in the savanna.<EOC><image> Output:`" and no support images would be fed to the model. We observed that only showing one, instead of two, text examples in the prompt is highly detrimental as the model is biased towards producing text output similar to the single provided text example. Providing more than two text examples helps but only marginally. We hence use two text examples in all zero-shot results for practicality. In practice, we believe this is not more cumbersome than finding a good natural text description for a given task. This relates to recent findings on the aspects of demonstrations that are key drivers of performance [76]. For close-ended tasks, where we use the model to score different possible answers, we observe it is not necessary to provide a single text example in the zero-shot prompt.

**Retrieval-based In-Context Example Selection [136].** When the size of the support set exceeds a certain limit, it can become difficult to leverage all the examples with in-context learning: first because it becomes excessively expensive to fit all the examples in the prompt, and second because there is a risk of poor generalization when the prompt size exceeds the size of the sequence used during training [83]. In such situations, it is appealing to use a form of prompt selection to both limit the sequence length as well as potentially improve the prompt quality which can in turn lead to better performance [63]. In particular, we follow the Retrieval-based In-Context Example Selection (RICES) approach introduced by [136]. In detail, given a query image, we retrieve similar images in the support set by comparing the visual features extracted from our frozen pretrained visual encoder. We then build the prompt by concatenating the top-$N$ most similar examples. Since LMs are sensitive to the ordering in the prompt due to recency bias [148], we order the examples by increasing order of similarity, such that the most similar support example appears right before the query. We notably show the effectiveness of this approach in classification settings with multiple hundreds of classes (see Appendix B.2.1) where we are given one or more images/videos per class, yielding a number of examples that would not otherwise fit in the prompt.

**Prompt ensembling.** We also explore ensembling the outputs of the model across multiple prompts in the close-ended setting. This can notably be combined with RICES where ensembling can be done over multiple permutations of the ranked nearest neighbors. Specifically, for a given answer, we average the log likelihoods estimated by the model over 6 random permutations of the selected few-shot examples.

## A.3 Training dataset details

We train the Flamingo models on a carefully chosen mixture of datasets illustrated in Figure 9 and described next.

### A.3.1 *M3W* collection

The selection and scraping of web pages for *M3W* follows a similar process to the one used for collecting the *MassiveWeb* dataset [86]. We start by filtering out non-English documents. We also remove those that do not pass internal filters, which identify explicit content across images, videos, and text. We use a custom scraper to extract salient content from the remaining documents, in the form of plain text interleaved with images, as described in Section 2.4. The text in *M3W* is collected in a similar fashion to that of *MassiveWeb*, but we also collect any images present at the same level in the HTML tree. We discard documents for which the scraping process does not yield any images.

We then apply similar text filtering heuristics, to remove low quality documents and reduce repetition, as well as some image filters to remove images that are too small (either width or height less than 64 pixels), too wide or narrow (aspect ratio greater than 3 in either direction), or unambiguously low quality (e.g. single-colour images). We discard documents that no longer contain any images following this filtering step.

### A.3.2 *M3W* image-placement augmentation

During evaluation of Flamingo models, we prompt the model with an image and ask it to generate text for that image. This lends itself to a natural sequencing at inference time in which the image comes before the corresponding text output.

However, the correspondence between images and text in our interleaved M3W dataset (Section 2.4) is in general unknown (and potentially not well-defined in certain cases). As a motivating example, a simple webpage might be structured in either of the following ways:

(a) This is my dog! <dog image>        This is my cat! <cat image>

(b) <dog image> That was my dog!        <cat image> That was my cat!

The text-aligned image indices (`indices`) might "ideally" be chosen such that at each point in the text, the index points to the most semantically relevant image for that text – i.e., the next image in example (a), and the previous image in example (b). In the absence of a general way to determine semantic correspondence between text and images on webpages "in the wild", we make a simplifying assumption that the most relevant image at any given point in the text is either the last image appearing before the text token, or the image immediately following it (as in the simple examples above), and choose `indices` accordingly.

During training, for each webpage sampled, we sample with probability $p_{next} = \frac{1}{2}$ whether `indices` are chosen to map text to the previous or next image. This inevitably means we make the semantically "unnatural" choice – e.g., associating the text "This is my cat!" with the dog image in (a) above – around half of the time. We ablate this choice in Section 3.3, finding a small advantage to setting $p_{next} = \frac{1}{2}$ over either 0 (always the previous image index) or 1 (always the next image index). This suggests that there may be a beneficial "data augmentation" effect to this randomisation.

### A.3.3 LTIP and VTP: Visual data paired with text

Along with our interleaved image and text dataset, we use several paired vision and text web datasets for training. One dataset is ALIGN [50], composed of 1.8 billion images paired with alt-text. ALIGN is large, but noisy and limited to images. The images are often poorly described by the corresponding alt-text annotation. For this reason, we augment it with two datasets: LTIP (Long Text & Image Pairs) consists of 312 million images, and VTP (Video & Text Pairs) consists of 27 million short videos (approximately 22 seconds on average). Both datasets are paired with more descriptive captions. For instance, the average number of tokens of an ALIGN text description is 12.4 per image, while it is 20.5 for the LTIP dataset. The LTIP and VTP datasets were collected by crawling fewer than ten websites targeting high-quality and rich image descriptions. These single-image and single-video datasets are preprocessed analogously to the *M3W* data preprocessing described previously, adding the <image> tag at the beginning of the sequence (immediately after <BOS>), and the <EOC> token after the text (before <EOS>). We deduplicated these datasets against all our benchmarks (against both the training and the evaluation sets) using image similarity, as detailed in Appendix A.3.4. Datasheets for LTIP and VTP are respectively given in Appendix F.2.1 and Appendix F.2.2.

| | Requires model sharding | Frozen | | Trainable | | Total count |
| | | Language | Vision | GATED XATTN-DENSE | Resampler | |
|---|---|---|---|---|---|---|
| *Flamingo*-3B | ✗ | 1.4B | 435M | 1.2B (every) | 194M | **3.2B** |
| *Flamingo*-9B | ✗ | 7.1B | 435M | 1.6B (every 4th) | 194M | **9.3B** |
| *Flamingo* | ✓ | 70B | 435M | 10B (every 7th) | 194M | **80B** |

Table 5: **Parameter counts for Flamingo models.** We focus on increasing the parameter count of the frozen LM and the trainable vision-text GATED XATTN-DENSE modules while maintaining the frozen vision encoder and trainable Resampler to a fixed and small size across the different models. The frequency of the GATED XATTN-DENSE with respect to the original language model blocks is given in parentheses.

### A.3.4 Dataset deduplication against evaluation tasks

We used an internal deduplication tool to deduplicate our training datasets from our evaluation datasets. This deduplication pipeline relies on a trained visual encoder which maps embedding closer together when they are potential duplicates. Once the image embeddings have been computed, a fast approximate nearest neighbor search is performed on the training images to retrieve duplicate candidates from the validation datasets. For the paired image-text dataset, we have deduplicated our LTIP and ALIGN training images against: ImageNet (train, val), COCO (train, valid, test), OK-VQA (train, valid, test), VQAv2 (train, valid, test), Flickr30k (train, valid, test), VisDial (train, valid, test).

We did not deduplicate our image datasets against VizWiz, HatefulMemes and TextVQA as we performed these evaluations only after having trained our Flamingo models. However, we believe this had no impact on our results as the images from these datasets are unlikely to be scraped from the web; VizWiz images were obtained using a specific mobile app and only available for download, HatefulMemes memes were created by researchers instead of being scraped on the web and finally TextVQA images are from OpenImages.

Note that we did not run the deduplication on the *M3W* dataset as one training example is a full webpage of interleaved paragraph with several images, unlikely to contain images from our benchmark suite. To verify this hypothesis, we have obtained near-duplicate statistics on the 185M individual images from *M3W* and the results are the following: in total, 1314 potential duplicates were found from the validation and test splits of ImageNet, COCO, OK-VQA, VQAv2, Flickr30k and VisDial. Out of the 1314 candidates, only 125 are exact duplicates.

For the video datasets, we did not perform any deduplication of VTP (27M videos) as none of the collected VTP videos were obtained from YouTube or Flickr, which are the sources of all of our video evaluation datasets collected on the Internet.

## B Experiments

### B.1 Training and evaluation details

#### B.1.1 Models

We perform experiments across three model sizes, where we scale the frozen language model from 1.4B to 7B and 70B; and adapt the parameter count of other components accordingly. We keep the pretrained vision encoder frozen across all experiments and use a NFNet-F6 model trained contrastively (see Appendix B.1.3), unless explicitly stated otherwise in the ablation study. We use a Perceiver Resampler with approximately 200M parameters across all three model sizes.

The decision on how many GATED XATTN-DENSE layers to interleave is mainly driven by a trade-off between memory constraints and downstream performance. We identified the optimal trade-off at small model scales, before transferring our findings to the large model architecture.

We obtain three models, *Flamingo*-3B, *Flamingo*-9B and *Flamingo*-80B, detailed below:

- The *Flamingo*-3B model builds on top of a **1.4B frozen language model** from [42]. Before each transformer block, we add a GATED XATTN-DENSE layer attending to the visual inputs; this accounts for 1.4B additional learned parameters.

- The *Flamingo*-9B model builds on top of a **7B frozen language model** from [42]. Starting from the very first layer and before every fourth transformer blocks, we add a GATED XATTN-DENSE layer attending to the visual inputs; this accounts for 1.8B additional learned parameters.

- The *Flamingo*-80B model builds on top of **the frozen Chinchilla 70B** language model [42]. Starting from the very first layer and before every seventh transformer blocks, we add a GATED XATTN-DENSE layer attending to the visual inputs; this accounts for 10B additional learned parameters. For simplicity, we refer to this model as simply *Flamingo* throughout the paper.

In Table 5 we report the parameter count of each component of our models, as well as model sharding requirements. We provide more Transformer architecture details in Appendix A.1.4. The *Flamingo* model card [77] is also given in Appendix E.

### B.1.2 Training details for the Flamingo models

**Data augmentation and preprocessing.** Empirically we find that it is effective to stochastically prepend the paired dataset text samples with a single space character, with probability 0.5. We attribute this to the fact that our subword tokenizer maps the beginning of various words to a different token depending on whether it is preceded by a space. This allows us to enforce invariance to this tokenizer artifact, without degrading significantly correctness of the punctuation which is already lacking in many of these samples. We observe that this leads to substantial improvement across tasks.

The visual inputs are resized to $320 \times 320$ while preserving their aspect ratios, padding the image with the mean value if required. Note that this is higher than the $288 \times 288$ resolution used for the contrastive pretraining of our Vision Encoder (see Appendix B.1.3). The increase in resolution during the final stage training was motivated by [113] showing one can obtain improved performance at a higher test-time resolution when using CNNs. This increase in resolution also comes with only a moderate computational and memory cost as no backpropagation is performed through the frozen Vision Encoder. We also employ random left/right flips and color augmentation.

For interleaved datasets (Section 2.4) we also employ augmentation by lightly randomizing the selected image indices $\phi$ with a hyperparameter $p_{next}$ when sampling examples from the *M3W* dataset. This augmentation is detailed in Appendix A.3.2 and our choice of $p_{next} = \frac{1}{2}$ is ablated in Appendix B.3.1. For video training, we temporally sample a clip of 8 frames sampled at one frame per second (fps) from each training video. Although our model was trained with a fixed number of 8 frames, at inference time, we input 30 frames at 3 FPS. This is achieved by linearly interpolating the learnt temporal position embedding of the Perceiver Resampler at inference time.

**Loss and optimisation.** All our models are trained using the AdamW optimizer with global norm clipping of 1, no weight decay for the Perceiver Resampler and weight decay of 0.1 for the other trainable parameters. The learning rate is increased linearly from 0 to $10^{-4}$ up over the first 5000 steps then held constant for the duration of training (no improvements were observed from decaying the learning rate). Unless specified otherwise we train our models for $500k$ steps. Four datasets are used for training: *M3W*, ALIGN, LTIP and VTP with weights $\lambda_m$ of 1.0, 0.2, 0.2 and 0.03 respectively. These weights were obtained empirically at a small model scale and kept fixed afterwards. Batch sizes depend on the setting and are given in the next sections.

**Infrastructure and implementation.** Our model and associated infrastructure were implemented using JAX [8] and Haiku [40]. All training and evaluation was performed on TPUv4 instances. The largest model containing 80 billion parameters is trained on 1536 chips for 15 days and sharded across 16 devices. Megatron type sharding [99] is used to enable 16-way model parallelism for all Embedding / Self-Attention / Cross-Attention / FFW layers, while the NFNet vision layers were unsharded. ZeRO stage 1 [88] is used to shard the optimizer state. All trained parameters and optimizer accumulators are stored and updated in `float32`; all activations and gradients are computed in `bfloat16` after downcasting of parameters from `float32` to `bfloat16`. Frozen parameters are stored and applied in `bfloat16`.

### B.1.3 Contrastive model details

The vision encoder is trained from scratch, together with a language encoder. Using these encoders, images and text pairs are separately encoded and projected to a shared embedding space and L2 normalized. From these embeddings, we maximize the similarity of paired embeddings and minimize the similarity of unpaired embeddings, using a multi-class cross-entropy loss, where the paired image-texts are treated as positive examples and the rest of the batch as negative examples. We use the same loss as in CLIP [85], which consists of two contrastive losses, one from text to image and the other from image to text. We use a learnable temperature parameter in the final log-softmax layer [9]. The text-to-image loss is as follows:

$$L_{contrastive:txt2im} = -\frac{1}{N} \sum_i^N \log \left( \frac{\exp(L_i^\intercal V_i \beta)}{\sum_j^N \exp(L_i^\intercal V_j \beta)} \right) \tag{3}$$

And the image-to-text loss is defined analogously:

$$L_{contrastive:im2txt} = -\frac{1}{N} \sum_i^N \log \left( \frac{\exp(V_i^\intercal L_i \beta)}{\sum_j^N \exp(V_i^\intercal L_j \beta)} \right) \tag{4}$$

The sum of the two losses is minimized. Here, $V_i$ and $L_i$ are, respectively, the normalized embedding of the vision and language component of the $i$-th element of a batch. $\beta$ is a trainable inverse temperature parameter and $N$ is the number of elements in the batch. We use the BERT [23] architecture for the language encoder. The outputs of the language and vision encoders are mean-pooled (across tokens and spatial locations, respectively) before being projected to the shared embedding space. We only use the weights from the contrastive vision encoder in the main Flamingo models.

The vision encoder is pretrained on the ALIGN and LTIP datasets. The training image resolution is $288 \times 288$, the joint embedding space is size 1376 and the batch size is 16,384. It is trained for 1.2 million parameter update steps, each of which consist of two gradient calculation steps (more details below) on 512 TPUv4 chips. The learning rate is decayed linearly from $10^{-3}$ to zero over the course of training. Images have random color augmentation and horizontal flips applied during training. We use the tokenizer employed by Jia et al. [50]. The Adam optimizer is used to optimize the network, and we apply label smoothing of 0.1. We apply $10^{-2}$ adaptive gradient clipping (AGC) [10] to the NFNet encoder and global norm gradient clipping of 10 for the BERT encoder.

To evaluate the pretrained model, we track zero-shot image classification and retrieval. For zero-shot image classification, we use image-text retrieval between the images and the class names. Following Radford et al. [85] we use "prompt-ensembling" in which we embed multiple texts using templates such as `"A photo of a {class_name}"` and average the resulting embedding.

### B.1.4 Evaluation benchmarks

Our goal is to develop models that can rapidly adapt to diverse and challenging tasks in the few-shot setting. For this, we consider a wide array of popular image and video benchmarks summarized in Table 6. In total we chose 16 multimodal image/video and language benchmarks, spanning tasks that require some language understanding (visual question answering, captioning, visual dialogue) as well as two standard image and video classification benchmarks (ImageNet and Kinetics). Note that for the video datasets collected from YouTube (i.e., all video datasets except NextQA and STAR), we evaluated our model on all the publicly available video as of April 2022.

**DEV benchmarks.** In order to validate design decisions of our model over the course of the project, we selected five benchmarks from the 16 multimodal image/video and language benchmarks as well as ImageNet and Kinetics for classification as our development set (referred as DEV). To maximise its relevance, we choose the most challenging and widely studied benchmarks for captioning, visual question-answering and classification tasks on both images and videos.

**Dataset splits for the DEV benchmarks.** Concretely, estimating few-shot learning performance of a model consists of adapting it on a set of *support* samples and evaluating it on a set of *query* samples. As a result, any evaluation set should be composed of two disjoint subsets containing respectively the support and the query samples. For the DEV benchmarks that are used both to validate design decisions and hyperparameters, as well as to report final performance, we therefore use four subsets:

| | Dataset | DEV | Gen. | Custom prompt | Task description | Eval set | Metric |
|---|---|---|---|---|---|---|---|
| **Image** | ImageNet-1k [94] | ✓ | | | Object classification | Val | Top-1 acc. |
| | MS-COCO [15] | ✓ | ✓ | | Scene description | Test | CIDEr |
| | VQAv2 [3] | ✓ | ✓ | | Scene understanding QA | Test-dev | VQA acc. [3] |
| | OKVQA [69] | ✓ | ✓ | | External knowledge QA | Val | VQA acc. [3] |
| | Flickr30k [139] | | ✓ | | Scene description | Test (Karpathy) | CIDEr |
| | VizWiz [35] | | ✓ | | Scene understanding QA | Test-dev | VQA acc. [3] |
| | TextVQA [100] | | ✓ | | Text reading QA | Val | VQA acc. [3] |
| | VisDial [20] | | | | Visual Dialogue | Val | NDCG |
| | HatefulMemes [54] | | | ✓ | Meme classification | Seen Test | ROC AUC |
| **Video** | Kinetics700 2020 [102] | ✓ | | | Action classification | Val | Top-1/5 avg |
| | VATEX [122] | ✓ | ✓ | | Event description | Test | CIDEr |
| | MSVDQA [130] | ✓ | ✓ | | Event understanding QA | Test | Top-1 acc. |
| | YouCook2 [149] | | ✓ | | Event description | Val | CIDEr |
| | MSRVTTQA [130] | | ✓ | | Event understanding QA | Test | Top-1 acc. |
| | iVQA [135] | | ✓ | | Event understanding QA | Test | iVQA acc. [135] |
| | RareAct [73] | | | ✓ | Composite action retrieval | Test | mWAP |
| | NextQA [129] | | ✓ | | Temporal/Causal QA | Test | WUPS |
| | STAR [128] | | | | Multiple-choice QA | Test | Top-1 acc. |

Table 6: **Summary of the evaluation benchmarks.** DEV benchmarks were used to validate general design decision of the Flamingo models. Gen. stands for generative task where we sample text from the VLM. If a task is non-generative it means that we use the VLM to score answers among a given finite set. For most of our tasks we use a common default prompt, hence minimizing task-specific tuning (see Appendix B.1.5).

- *validation support*: contains support samples for validation;

- *validation query*: contains query samples for validation;

- *test support*: contains support samples for final performance estimation;

- *test query*: contains query samples for final performance estimation.

In practice, for the *test query* subset, we use the subset that prior works report results on, for apples-to-apples comparison. While the validation set would be a natural choice for the *validation query* subset, we note that this is not possible for all benchmarks, since some benchmarks do not have an official validation set (e.g. OKVQA) and for others, the validation is commonly used to report final performance in place of the test set (e.g. ImageNet or COCO). For simplicity, we use a subset of the original training set as the *validation query* subset. Finally, we also use additional disjoint subsets of the training set as respectively the *validation support* subset and the *test support* subset.

We now describe in more detail how we form the latter three subsets. For captioning tasks, open-ended evaluation is efficient so we evaluate on a large number of samples. Specifically, for COCO, we use the same number of samples as used in the Karpathy splits for evaluation sets (5000). For VATEX, because the training set is of limited size, we only evaluate over 1024 samples, reserving the rest for support sets. For question-answering tasks, we evaluate over 1024 samples; chosen to make both open- and close-ended evaluation reasonably fast. For image classification tasks, we evaluate over 10 images per class: 10,000 samples for ImageNet, and 7000 samples for Kinetics700. As for the support sets, for both validation and final performance estimation, we use 2048 samples across all tasks, except for classification tasks where we scale this to 32 samples per class, to better estimate expected performance for each class.

**Unbiased few-shot performance estimation.** Few-shot learning performance estimates on the DEV benchmarks may be biased, in the sense that over the course of this project, design decisions were made based on the performance obtained on these benchmarks. We note that this is the case for prior work which also make use of these benchmarks to validate and ablate their own design decisions. To account for this bias and provide unbiased few-shot learning performance estimates, we report performance on a remaining set of 11 benchmarks. Among those, some span the same open-ended image and video tasks as our DEV benchmarks (captioning and visual question-answering). But we also look at more specific benchmarks in order to explore less explored capabilities. These notably include: TextVQA [100] which specifically assesses OCR capabilities through question-answering;

VisDial [20], a visual dialogue benchmark; HatefulMemes [54] a vision and text classification benchmark; NextQA [129] which specially focuses on causality and temporal relation; STAR [128], a multiple-choice question answering task; and RareAct [73], a benchmark measuring compositionality in action recognition. We emphasize that *we do not validate any design decisions* on these benchmarks and use them solely to estimate unbiased few-shot learning performance after Flamingo training is done.

### B.1.5 Few-shot learning evaluation hyperparameters

In few-shot learning, hyperparameter selection implicitly increases the number of shots as it requires additional validation examples. If those are not taken into account, as is often the case in practice, few-shot performance can be overestimated [80]. Similarly, cross-validation of benchmark-specific hyperparameters such as the prompt should be considered as a particularly basic few-shot learning method, where one selects the task-specific prompt over the set of shots. But other learning approaches might be more effective in making use of these labelled examples. Given the negative results reported by [80] in terms of the robustness of cross-validation and unless mentioned otherwise, all benchmarks are run using a single set of evaluation hyperparameters, including the prompts. We optimize hyperparameters jointly across the *validation* subsets of the DEV benchmarks and do not perform any benchmark-specific cross-validation of hyperparameters, aside from a few exceptions, as we detail next.

Except for HatefulMemes and RareAct, we always use the prompt "'`Output: {output}`" for all non-question-answering tasks, and "`Question: {question} Answer: {answer}`" for all question-answering / visual dialogue tasks. In particular, for VisDial [20], we use the previously described prompt to encode each questions/answers in the dialogue and the provided image caption is prepended to the dialogue history without any prompt. For HatefulMemes [54], we use a specific prompt to incorporate the OCR information provided as input which is: "`is an image with written: "{meme_text}" on it.  Is it hateful? Answer: {answer}`", where the answer is either yes or no. Note that this is the only dataset where we explicitly provide OCR text "`meme_text`" as input to Flamingo models. For TextVQA, we do not make use of the provided OCR transcripts and instead directly rely on the off-the-shelf OCR capabilities of the Flamingo models. For RareAct, a zero-shot benchmark, we change the verb names to the third person, add an article before each noun and use the prompt "`Caption:  a person {verb + object}`".

### B.1.6 Dialogue prompt

For the dialogue examples in Figures 1 and 11, the following prompt is used:

```
This is a conversation between a human, User, and an intelligent
visual AI, Flamingo.  User sends images, and Flamingo describes
them.

User:  <a cat image>

Flamingo:  That is a cat.  It's a tiny kitten with really cute big
ears.

User:  <a dinner image>

Flamingo:  This is a picture of a group of people having dinner.
They are having a great time!

User:  Can you guess what are they celebrating?

Flamingo:  They might be celebrating the end of a successful
project or maybe a birthday?

User:  <a graph image>

Flamingo:  This is a graph, it looks like a cumulative density
function graph.
```

| Model | Method | Prompt size | shots/class | ImageNet top 1 | Kinetics700 avg top1/5 |
|---|---|---|---|---|---|
| SotA | Fine-tuned | - | full | 90.9 [127] | 89.0 [134] |
| SotA | Contrastive | - | 0 | **85.7 [82]** | **69.6 [85]** |
| NFNetF6 | Our contrastive | - | 0 | 77.9 | 62.9 |
| *Flamingo*-3B | RICES | 8 | 1 | 70.9 | 55.9 |
| | | 16 | 1 | 71.0 | 56.9 |
| | | 16 | 5 | 72.7 | 58.3 |
| *Flamingo*-9B | RICES | 8 | 1 | 71.2 | 58.0 |
| | | 16 | 1 | 71.7 | 59.4 |
| | | 16 | 5 | 75.2 | 60.9 |
| *Flamingo*-80B | Random | 16 | ≤ 0.02 | 66.4 | 51.2 |
| | RICES | 8 | 1 | 71.9 | 60.4 |
| | | 16 | 1 | 71.7 | 62.7 |
| | | 16 | 5 | 76.0 | 63.5 |
| | RICES+ensembling | 16 | 5 | 77.3 | 64.2 |

Table 7: **Few-shot results on classification tasks.** The Flamingo models can also be used for standard classification tasks. In particular, we explore having access to support sets bigger than what our current prompt can accommodate (using up to 5000 support examples). In that regime, large gains are obtained by using the RICES method [136] as well as prompt ensembling. We also observe the same trend as with the vision-language benchmarks: bigger models do better and more shots help.

## B.2   Additional performance results

### B.2.1   Few-shot learning on classification tasks

We consider applying the Flamingo models to well-studied classification benchmarks like ImageNet or Kinetics700. Results are given in Table 7. We observe a similar pattern as in other experiments: larger model tend to perform better. Second, given that few-shot classification tasks often come with more training examples (e.g., 1000 for ImageNet with 1 example per class), using methods to scale to larger support sets is beneficial. RICES (Retrieval In-Context Example Selection [136] described in Appendix A.2) performs substantially better than simply selecting examples randomly for inclusion in the prompt. Indeed, *Flamingo* achieves a 9.2% improvement in ImageNet classification when selecting 16 support examples out of 5000 using RICES, compared to choosing the same number of examples randomly. Ensembling multiple prompts further boosts results. However, note that Flamingo models underperform the current dominant contrastive paradigm for classification tasks; in particular, they underperform the very contrastive model used as their vision encoder (see Appendix D.1 on Flamingo's limitations for more details). Finally, state-of-the-art zero-shot models on ImageNet such as BASIC [82] and LiT [146] are particularly optimized on classification tasks as they are trained on JFT-3B [145], a dataset with images and labels. Improving the performance of VLMs such as Flamingo on classification tasks is an interesting direction for future work.

### B.2.2   Fine-tuning *Flamingo* as a pretrained vision-language model

To fine-tune Flamingo models on a downstream task, we train them on data batches from the task of interest in the same format as the single-image/video datasets described in Section 2.4.

**Freezing and hyperparameters.**   When fine-tuning *Flamingo*, we keep the underlying LM layers frozen and train the same Flamingo layers as during pretraining. We also increase the resolution of the input images from $320 \times 320$ to $480 \times 480$. Unlike in the pretraining phase, we also fine-tune the base visual encoder, finding that this typically improves results, likely due in part to the higher input resolution.

We choose certain hyperparameters on a per-task basis by grid search on a validation subset of the training set (or on the official or standard validation set where available). These hyperparameters include the learning rate (ranging from $3 \times 10^{-8}$ to $1 \times 10^{-5}$) and decay schedule (exponential decay

by factors of $10\times$), number of training steps, batch size (either 8 or 16), and whether visual data augmentation (color augmentation, random horizontal flips) is used.

**Results.** In Table 8, we present additional results for per-task *Flamingo* fine-tuning. When provided access to a large-scale task-specific dataset with many thousands of examples, we find that we can improve results over our previously presented in-context few-shot learning results, setting a new state of the art on five tasks: VQAv2, VATEX, VizWiz, MSRVTTQA, and HatefulMemes. For example, on VQAv2, we observe improved results at $82.0\%$, outperforming our results achieved with 32-shot in-context learning ($67.3\%$) as well as the previous state of the art ($81.3\%$, Yan et al. [133]).

Although these fine-tuning results come at high computational cost relative to the previously presented in-context few-shot learning results – among other challenges like hyperparameter tuning – they further demonstrate the power of VLM pretraining for visual understanding even in the presence of large amounts of task-specific training data.

In some cases our results likely trail the state of the art due in part to the fact that we simply optimise log-likelihood and do not make use of common task-specific metric optimisation tricks, such as CIDEr optimisation [64, 90] for COCO captioning, and fine-tuning on dense annotations for VisDial [79]. For example, Murahari et al. [79] report a $10\%$ relative improvement in NDCG on VisDial from such dense annotation fine-tuning.

### B.2.3 Zero-shot performance of the pretrained contrastive model

A crucial part of our approach is the Vision Encoder, pretrained separately using contrastive learning and kept frozen when training Flamingo models. We report zero-shot image classification results on ImageNet, Kinetics700 and retrieval results on Flick30K and COCO. The classification results are presented in Table 7 while the retrieval results are given in Table 9. For the retrieval tasks, our model outperforms the current state-of-the-art contrastive dual encoder approaches CLIP [85], ALIGN [50] and Florence [140]. However, we underperform the zero-shot state-of-the-art on Kinetics700 (CLIP) and the zero-shot state-of-the-art on ImageNet (BASIC). However, as noted earlier, BASIC [82] is particularly optimized for classification: it is trained on the JFT-3B [145] dataset which has images with labels rather than captions. We have noticed training on image and short text descriptions similar to labels significantly helps for ImageNet but is detrimental for retrieval benchmarks which require capturing rich scene descriptions instead. Since our goal is to use the Vision Encoder as a feature extractor for the Flamingo models in order to capture the whole scene and not just the main object, we favor retrieval metrics over classification ones. We provide more details about the contrastive pretraining in Appendix B.1.3.

Table 8: **Comparison to SotA when fine-tuning *Flamingo*.** We fine-tune *Flamingo* on all nine tasks where *Flamingo* was not SotA with few-shot learning. *Flamingo* sets a new SotA on five of these tasks sometimes even beating methods that resorts to known performance optimization tricks such as model ensembling (on VQAv2, VATEX, VizWiz and HatefulMemes). Best numbers among the restricted SotA are in **bold**. Best numbers overall are underlined. Restricted SotA† only includes methods that use a single model (not ensembles) and do not directly optimise the test metric (no CIDEr optimisation).

| Method | VQAV2 | | COCO | VATEX | VizWiz | | MSRVTTQA | VisDial | | YouCook2 | TextVQA | | HatefulMemes |
| --- | --- | --- | --- | --- | --- | --- | --- | --- | --- | --- | --- | --- | --- |
| | test-dev | test-std | test | test | test-dev | test-std | test | valid | test-std | valid | valid | test-std | test seen |
| 🦩 *Flamingo* - 32 shots | 67.6 | - | 113.8 | 65.1 | 49.8 | - | 31.0 | 56.8 | - | 86.8 | 36.0 | - | 70.0 |
| SimVLM [124] | 80.0 | 80.3 | **143.3** | - | - | - | - | - | - | - | - | - | - |
| OFA [119] | 79.9 | 80.0 | 149.6 | - | - | - | - | - | - | - | - | - | - |
| Florence [140] | 80.2 | 80.4 | - | - | - | - | - | - | - | - | - | - | - |
| 🦩 *Flamingo* Fine-tuned | **82.0** | **82.1** | 138.1 | **84.2** | **65.7** | **65.4** | **47.4** | 61.8 | 59.7 | 118.6 | **57.1** | 54.1 | **86.6** |
| Restricted SotA† | 80.2 | 80.4 | **143.3** | 76.3 | - | - | 46.8 | **75.2** | 74.5 | **138.7** | 54.7 | **73.7** | 79.1 |
| | [140] | [140] | [124] | [153] | - | - | [51] | [79] | [79] | [132] | [137] | [84] | [62] |
| Unrestricted SotA | 81.3 | 81.3 | 149.6 | 81.4 | 57.2 | 60.6 | - | - | 75.4 | - | - | - | 84.6 |
| | [133] | [133] | [119] | [153] | [65] | [65] | - | - | [123] | - | - | - | [152] |

| | Flickr30K | | | | | | COCO | | | | | |
|---|---|---|---|---|---|---|---|---|---|---|---|---|
| | image-to-text | | | text-to-image | | | image-to-text | | | text-to-image | | |
| | R@1 | R@5 | R@10 | R@1 | R@5 | R@10 | R@1 | R@5 | R@10 | R@1 | R@5 | R@10 |
| Florence [140] | **90.9** | **99.1** | - | 76.7 | 93.6 | - | 64.7 | 85.9 | - | 47.2 | 71.4 | - |
| ALIGN [50] | 88.6 | 98.7 | **99.7** | 75.7 | 93.8 | 96.8 | 58.6 | 83.0 | 89.7 | 45.6 | 69.8 | 78.6 |
| CLIP [85] | 88.0 | 98.7 | 99.4 | 68.7 | 90.6 | 95.2 | 58.4 | 81.5 | 88.1 | 37.7 | 62.4 | 72.2 |
| **Ours** | 89.3 | 98.8 | **99.7** | **79.5** | **95.3** | **97.9** | **65.9** | **87.3** | **92.9** | **48.0** | **73.3** | **82.1** |

Table 9: **Zero-shot contrastive pretraining evaluation.** Zero-shot image-text retrieval evaluation of our pretrained contrastive model compared to the state-of-the-art dual encoder contrastive models.

| | Ablated setting | Flamingo 3B value | Changed value | Param. count ↓ | Step time ↓ | COCO CIDEr↑ | OKVQA top1↑ | VQAv2 top1↑ | MSVDQA top1↑ | VATEX CIDEr↑ | Overall score↑ |
|---|---|---|---|---|---|---|---|---|---|---|---|
| | **Flamingo 3B model (short training)** | | | 3.2B | 1.74s | 86.5 | 42.1 | 55.8 | 36.3 | 53.4 | **70.7** |
| **(i)** | Resampler size | Medium | Small | 3.1B | 1.58s | 81.1 | 40.4 | 54.1 | 36.0 | 50.2 | 67.9 |
| | | | Large | 3.4B | 1.87s | 84.4 | 42.2 | 54.4 | 35.1 | 51.4 | 69.0 |
| **(ii)** | Multi-Img att. | Only last | All previous | 3.2B | 1.74s | 70.0 | 40.9 | 52.0 | 32.1 | 46.8 | 63.5 |
| **(iii)** | $p_{next}$ | 0.5 | 0.0 | 3.2B | 1.74s | 85.0 | 41.6 | 55.2 | 36.7 | 50.6 | 69.6 |
| | | | 1.0 | 3.2B | 1.74s | 81.3 | 43.3 | 55.6 | 36.8 | 52.7 | 70.4 |
| **(iv)** | LM pretraining | MassiveText | C4 | 3.2B | 1.74s | 81.3 | 34.4 | 47.1 | 60.6 | 53.9 | 62.8 |
| **(v)** | Freezing Vision | ✓ | ✗ (random init) | 3.2B | 4.70s* | 74.5 | 41.6 | 52.7 | 31.4 | 35.8 | 61.4 |
| | | | ✗ (pretrained) | 3.2B | 4.70s* | 83.5 | 40.6 | 55.1 | 34.6 | 50.7 | 68.1 |
| **(vi)** | Co-train LM on MassiveText | ✗ | ✓ (random init) | 3.2B | 5.34s* | 69.3 | 29.9 | 46.1 | 28.1 | 45.5 | 55.9 |
| | | | ✓ (pretrained) | 3.2B | 5.34s* | 83.0 | 42.5 | 53.3 | 35.1 | 51.1 | 68.6 |
| **(vii)** | Dataset and Vision encoder | M3W+ITP+VTP and NFNetF6 | LAION400M and CLIP | 3.1B | 0.86s | 61.4 | 37.9 | 50.9 | 27.9 | 29.7 | 54.7 |
| | | | M3W+LAION400M+VTP and CLIP | 3.1B | 1.58s | 76.3 | 41.5 | 53.4 | 32.5 | 46.1 | 64.9 |

Table 10: **Additional ablation studies.** Each row in this ablation study table should be compared to the baseline Flamingo run reported at the top of the table. The step time measures the time spent to perform gradient updates on all training datasets. (*): Due to higher memory usage, these models were trained using four times more TPU chips. The obtained accumulation step time was therefore multiplied by four.

## B.3 Extended ablation studies

### B.3.1 Flamingo

**Ablation study experimental setup.** As in Table 10, we report per-task results and the Overall score (see Section 3.3) for *Flamingo*-3B on the *validation* subsets of the 5 DEV multimodal benchmarks with 4 shots in Table 10. We perform the ablation using batch size of 256 for *M3W*, 512 for ALIGN, 512 for LTIP and 64 for VTP. Models are trained for 1 million gradient steps (meaning 250,000 gradient updates, for the base model as we accumulate gradients over four datasets).

**Resampler size.** We further investigate the architectural design of the Resampler in row **(i)** of Table 10. We ablate the size of our Resampler with three options: Small, Medium (default value for all Flamingo models), and Large. We see that the best performance is achieved with a medium size Resampler. Moreover, when scaled together with the frozen LM, we observed that increasing the size of the Perceiver Resampler lead to unstable training. We thus made a conservative choice to keep the same medium Resampler size for all our Flamingo models.

**Effect of how many images are cross-attended to.** In the interleaved image-text scenario, we ablate whether the model can only attend to the single most recent previous image, or to all the previous images (row **(ii)** of Table 10). We can see that the single image case leads to significantly better results (7.2% better in the overall score). One potential explanation is that when attending to all previous images, there is no explicit way of disambiguating between different images in the cross-attention inputs. Nonetheless, recent work has shown that such disambiguation is still possible implicitly through the causal attention mechanism [36]. We also explored more explicit ways to enable this while attending to all previous images by modifying the image tags to include an index (<image 1>, <image 2>, etc.) and/or learning absolute index embeddings added to the cross-attention features for each image. These strategies were not as robust as our method when the number of images per sequence changes between training and test time. Such a property is desirable

to reduce the number of images per sequence during training for better efficiency (we use $N = 5$ at training time) while still generalizing to many images for few-shot evaluation (we go up to $N = 32$ at test time). For these reasons, we keep the single image cross-attention strategy for the Flamingo models. Note that while the model cannot explicitly attend to all previous images due to this masking strategy, it can still implicitly attend to them from the language-only self-attention that propagates all previous images' features via the previous text tokens.

*M3W* **image placement data augmentation.** Given a webpage, we don't know in advance if the text of the page will mention the previous or the next image in the two-dimensional layout of the page DOM. For this reason, we explore a data augmentation on *M3W* controlled by $p_{next}$ which indicates whether a given text token attends to the previous or the next image (see more details in Appendix A.3.2). The default value $p_{next} = \frac{1}{2}$ means that for each webpage sampled, we decide uniformly at random whether the model attends to the previous or next image. $p_{next} = 0$ means the model always attends to the previous image while $p_{next} = 1$ means the model always attends to the following image. The results (row **(iii)** of Table 10) show that using this randomization is beneficial.

**Language model pretraining.** To measure the importance of text pretraining, we compare the performance of using a frozen decoder-only Transformer either pretrained on MassiveText (our main model) or pretrained on the C4 dataset [87] (row **(iv)** of Table 10). Using the C4 dataset (which is smaller and less filtered than MassiveText) for training leads to a significant loss in performance ($-7.9\%$ overall). We note that the performance notably decreases for tasks that involve more language understanding such as visual question-answering tasks (OKVQA, VQAv2 and MSVDQA) while it remains on par for tasks that do not require as much language understanding (COCO, VATEX). This highlights the importance of pretraining the LM on a high-quality text-only dataset.

**Freezing the vision encoder.** During Flamingo training, we freeze the pretrained components (Vision Encoder and LM layers) while training newly added components from scratch. We ablate in **(v)** of Table 10 this freezing decision by training the Vision Encoder weights either from scratch or initialized with the contrastive vision-language task. If trained from scratch, we observe that the performance decreases by a large margin of $-9.3\%$. Starting from pretrained weights still leads to a drop in performance of $-2.6\%$ while also increasing the compute cost of the training.

**Alternative to freezing the LM by co-training on MassiveText.** Another approach for preventing catastrophic forgetting is to co-train on MassiveText [86], the dataset that was used to pretrain the language model. Specifically, we add MassiveText to the training mixture, with a weight $\lambda_m$ of $1.0$ (best performing after a small grid search), using a sequence length of $2048$ and the exact same setting as the pretraining of Chinchilla [42] for computing the text-only training loss. In order to co-train on MassiveText, we need to unfreeze the language model but we keep the vision encoder frozen. We perform two ablations in row **(vi)** of Table 10: starting from a pretrained language model (with a learning rate multiplier of $0.1$ of the LM weights) versus initializing from scratch (with the same learning rate everywhere). In both cases, the overall scores are worse than our baseline which starts from the language model, pretrained on MassiveText, and is kept frozen throughout training. This indicates that the strategy of freezing the language model to avoid catastrophic forgetting is beneficial. Even more importantly, freezing the LM is computationally cheaper as no gradient updates of the LM weights are required and we do not need to train on an additional dataset. This computational argument is even more relevant for our largest model, *Flamingo*-80B, where we freeze almost $90\%$ of the overall weights.

**Additional experiments using the LAION400M dataset.** In order to provide reference numbers that are more easily reproducible using publicly available datasets and network weights we also provide two additional ablations using the CLIP ViT L-14 weights [85] and the LAION400M dataset [96] in rows **(vii)** of Table 10.

### B.3.2 Dataset mixing strategies for the contrastive pretraining

One key to achieving strong results was the inclusion of our new dataset LTIP alongside ALIGN for training. Despite being a smaller dataset ALIGN by a factor of 6, a contrastive model trained on only LTIP outperforms one trained only on ALIGN on our evaluation metrics, suggesting that dataset quality may be more important than scale in the regimes in which we operate. We also find that a

| Dataset | Combination strategy | ImageNet accuracy top-1 | COCO | | | | | |
|---------|---------------------|-------------------------|------|------|------|------|------|------|
| | | | image-to-text | | | text-to-image | | |
| | | | R@1 | R@5 | R@10 | R@1 | R@5 | R@10 |
| LTIP | None | 40.8 | 38.6 | 66.4 | 76.4 | 31.1 | 57.4 | 68.4 |
| ALIGN | None | 35.2 | 32.2 | 58.9 | 70.6 | 23.7 | 47.7 | 59.4 |
| LTIP + ALIGN | Accumulation | **45.6** | **42.3** | **68.3** | **78.4** | **31.5** | **58.3** | **69.0** |
| LTIP + ALIGN | Data merged | 38.6 | 36.9 | 65.8 | 76.5 | 15.2 | 40.8 | 55.7 |
| LTIP + ALIGN | Round-robin | 41.2 | 40.1 | 66.7 | 77.6 | 29.2 | 55.1 | 66.6 |

Table 11: **Effect of contrastive pretraining datasets and combination strategies.** The first two rows show the effect of training a small model on LTIP and ALIGN only; the final three show the results of a small model trained on combinations of these datasets, comparing different combination strategies.

model trained on both ALIGN and LTIP outperforms those trained on the two datasets individually and that how the datasets are combined is important.

To demonstrate this, we train a small model with an NFNet-F0 vision encoder, BERT-mini language encoder and batch size 2048 for 1 million gradient-calculation steps on ALIGN, LTIP and a mixture of the two. The results are presented in Table 11. It shows the results of training models on the combined datasets using three different merging regimes:

- Data merged: Batches are constructed by merging examples from each dataset into one batch.
- Round-robin: We alternate batches of each dataset, updating the parameters on each batch.
- Accumulation: We compute a gradient on a batch from each dataset. These gradients are then weighted and summed and use to update the parameters.

Across all evaluation metrics, we find that the Accumulation method outperforms other methods of combining the datasets. Although the LTIP dataset is $5 \times$ smaller than the ALIGN dataset, this ablation study suggests that the quality of the training data can be more important than its abundance.

## C   Qualitative results

In addition to the samples in Figure 1, in this section we provide selected samples covering different interaction modalities in Figures 10, 11, and 12. Unlike the quantitative benchmark results which use beam search with a beam width of 3 for decoding, all qualitative results presented in this section use greedy decoding for faster sampling.

Figure 10 shows the simplest form of interaction where a single image is provided followed by a text prompt either in the form of a question or the start of a caption. Even though the model is not trained specifically for the question and answer format, the capabilities of the pretrained language model allows this adaptation. In many of these examples, *Flamingo* can do at least one step of implicit inference. Some of the objects are not named in the prompt but their properties are queried directly. Based on its visual input, the model manages to recall the knowledge relevant to the referred object and thus produces the correct answer. Vision networks trained contrastively have been shown to learn character recognition capabilities [85]. We observe that *Flamingo* preserves this capability in the full model, in some cases for text that is rather small with respect to the size of the image.

Since our model can accept inputs in the form of arbitrary sequences of visuals and language, we test its abilities to hold an extended dialogue with interleaved images and text. Figure 11 shows some samples which are generated by prompting the model with a brief dialogue (Appendix B.1.6) followed by user interaction including image insertions. Even after several rounds of interaction *Flamingo* can still successfully attend to the image and reply to questions that can not be guessed by language alone. We observe that multiple images can be separately attended: simple comparisons and inferences are handled properly.

Lastly, we investigated similar capabilities with video inputs as they present some extra challenges compared to images. Figure 12 shows some selected samples. As seen in the figure, in some cases

*Flamingo* can successfully integrate information from multiple frames (e.g., videos scanning through a scene or text) and answer questions involving temporal understanding (e.g., in the last example, with the word "after").

## D  Discussion

### D.1  Limitations, failure cases and opportunities

Here, we describe some limitations and failure cases of our models, as well as opportunities for further improving our models and extending their abilities.

**Classification performance.**  Although our visual language models have important advantages over contrastive models (e.g., few-shot learning and open-ended generation capabilities), their performance lags behind that of contrastive models on classification tasks. We believe this is because the contrastive training objective directly optimizes for text-image retrieval, and in practice, the evaluation procedure for classification can be thought of as a special case of image-to-text retrieval [85]. This is not the case for the language modeling objective we use to train our visual language models and this may contribute to the observed performance gap on classification tasks. In particular, Zhao et al. [148] have shown that language models suffer from various biases arising from the training data distribution, the set of samples used in the prompt, and their order. They also show that such issues can be mitigated with calibration techniques, provided one can assume a certain prior distribution (e.g., uniform) over the label space. This assumption doesn't hold in general, and further research is needed to develop techniques to address these issues in the few-shot setting. More generally, seeking objectives, architectures, or evaluation procedures that could bridge the gap between these two classes of models is a promising research direction.

**Legacies of language models.**  Our models build on powerful pretrained causal language models, and as a side effect, directly inherit their weaknesses. For instance, causal modeling of the conditioning inputs is strictly less expressive than bidirectional modeling. In this direction, recent work has shown that non-causal masked language modeling adaptation [120] followed by multitask fine-tuning [95, 125, 131] can efficiently improve the zero-shot performance of causal decoder-only language models. Furthermore, transformer-based language models tend to generalize poorly to test sequences significantly longer than the training ones [83]. In settings where the expected text output is too long, the ability of the models to leverage enough shots for few-shot learning can be affected. For instance, for the VisDial dataset [20], a single shot consists of an image followed by a long dialogue composed of 21 different sentences. A sequence of 32 VisDial shots is thus composed of at least $32 \times 21 = 672$ sentences, which in practice means that the prompt length ranges from 4096 to 8192 tokens. This is significantly longer than the maximum sequence length (2048) our LMs have been trained on [42]. To this end, we have capped our reported results on VisDial at 16 shots. On another note, while our ablations demonstrate the importance of the language model priors inherited from frozen language models, we suspect that they may play a role in occasional hallucinations and ungrounded guesses observed in open-ended dialogue settings. We provide and analyze examples of such behaviours in Figure 13. Finally, language modeling suffers from poor sample efficiency during pretraining [11]. Mitigating this issue has the potential to greatly accelerate progress in the field, by improving turnaround of large-scale training runs and in turn increasing the feasibility of more systematic exploration of design decisions at larger scales. Further discussion on typical weaknesses observed for large LMs can be found in [11, 86].

**Trade-offs of few-shot learning methods.**  In the paper, we use in-context learning as our "go-to" few-shot learning method (see Section 2.5). This method has notable advantages over gradient-based approaches such as fine-tuning. Indeed, in-context learning requires almost no hyperparameter tuning, works reasonably well in the very low data regime (dozens of examples), and only requires inference, simplifying deployment. In contrast, gradient-based approaches require carefully tuned design choices to avoid overfitting (either by proper learning rate schedule or architecture design [43]) and often need more data (thousands) to work well. This motivated our focus on in-context learning; however, this approach also has drawbacks we discuss next.

*Inference compute cost.* The compute cost of in-context learning with transformer models scales linearly with the number of shots if one can reuse the few-shot prompt for multiple query samples

[Figure]

Figure 10: **Selected single image samples.** Gray boxes are user input and the pink boxes are *Flamingo* output.

[Figure]

This is a dog. It's a white fluffy dog.

What season is this?

It looks like a summer day.

Why do you think so?

The grass is green and the dog is not wearing a sweater.

What breed is the dog?

It's a Samoyed.

Is the dog running?

No, it's sitting.

Can you describe the pose of its back legs?

The back legs are bent and the dog is sitting on its haunches.

[Figure]

Which city is this?

This is a street sign in New York City.

Which street corner is it?

It is the intersection of Madison Avenue and East 42nd Street.

Which subway station is nearby?

The closest subway station is Grand Central Station.

How about the nearest park?

The nearest park is Bryant Park.

There is another sign. What does it say?

It says "One way".

[Figure]

I would like you to read the color and then say the color it is written in. For example:

PURPLE

Color is "Purple" and it is written in red.

GREEN

Color is "Green" and it is written in blue.

YELLOW

Color is "Yellow" and it is written in green.

BLACK

Color is "Black" and it is written in yellow.

Well done! Do you know the name of the test these images come from?

I think it is called the Stroop test.

Can you explain how humans perform in this test?

Humans are slower when the color of the word and the color of the word are different.

How about you?

I am not affected by this difference.

[Figure]

What is the difference between these two images?

The first one is a triangle and the second one is a hexagon.

Which one has larger internal angles?

The hexagon has larger internal angles.

What color are they?

They are both red.

[Figure]

What is in this picture?

It's a bowl of soup with a monster face on it.

What is the monster made out of?

It's made out of vegetables.

No, it's made out of a kind of fabric. Can you see what kind?

It's made out of a woolen fabric.

[Figure]

This is a picture of a dog. It looks like it is a little bit naughty.

What happened here?

It looks like the dog is destroying the sofa.

How does the person thinking about it?

The person is angry.

What is the person holding?

It looks like a cup of coffee.

How is the weather outside?

It looks like a sunny day.

[Figure]

What is the name of the company on this card?

It's Homebase.

What is its web url?

It's homebase.co.uk.

What are they doing?

They are a home improvement retailer.

[Figure]

What do you think the robot is doing here?

The robot is picking up a ball.

What color is the ball?

It's red.

I want the ball to be hidden. Is there something the robot can use for that?

The robot can use a cloth to hide the ball.

What color is the cloth?

The cloth is blue.

Figure 11: **Selected dialogue samples.** Gray boxes are user input and the pink boxes are *Flamingo* output. For dialogue, *Flamingo* is provided with a custom prompt (hidden from the visualization but shown in Appendix B.1.6) containing a dialogue with 3 corresponding images, but it is not fine-tuned for dialogue in any other way.

[Figure]

Figure 12: **Selected video samples.** These are all of the frames the model sees. (Best viewed with zoom.)

(by caching the keys and values) and quadratically otherwise. In contrast, gradient-based few-shot learning approaches [43] have constant complexity with respect to the number of shots during inference.

*Prompt sensitivity.* In-context learning has also been shown to be disconcertingly sensitive to various aspects of the demonstrations, such as the order of the samples [148] or their format.

*Leveraging more shots.* When using in-context learning, performance plateaus rapidly as the number of few-shot samples increases beyond 32. This proves a striking contrast with typical gradient-based methods, for which the amount of correctly paired training data is a critical factor for performance. We note that RICES (Retrieval In-Context Example Selection [136] described in Appendix A.2) effectively mitigates this issue for classification tasks (Appendix B.2.1), but still faces similar issues beyond a small number of example per class.

*Task location.* Recent work on understanding what makes in-context learning effective sheds some light on a possible explanation for why more shots do not always help [76, 92]. In more detail, Brown et al. [11] raise the question of whether in-context learning actually "learns" new tasks at inference time based on the provided input-output mappings, or simply recognizes and identifies tasks learned during training. On this question, the findings of Reynolds and McDonell [92] suggest that the latter is the key driver of performance across diverse settings, and refer it as *task location*. Similarly, Min et al. [76] show that the mapping from input to output generally has limited impact on few-shot performance, as opposed to specifying the overall format of the examples. In line with these findings, we also observe non-trivial zero-shot performance using prompt without any images, hence also highlighting that the format of the task matters significantly. Intuitively, a handful of samples may often be enough to perform task location well, but the model may generally not be able to leverage further samples at inference time to refine its behaviour.

[Figure]

Figure 13: **Hallucinations and ungrounded guesses in open-ended visual question answering.** *Left:* The model occasionally hallucinates by producing answers that seem likely given the text only, but are wrong given the image as additional input. *Middle:* Similar hallucinations can be provoked by adversarially prompting the model with an irrelevant question. *Right:* A more common pitfall arises when the model makes ungrounded guesses when the answer cannot be determined based on the inputs. Few-shot examples and more sophisticated prompt design may be used to mitigate these issues. More broadly, addressing these issues is an important research direction towards improving our models' applications in open-ended visual dialogue settings.

In summary, there is no "golden" few-shot method that would work well in all scenarios. In particular, the best choice of few-shot learning approach strongly depends on characteristics of the application, an important one being the number of annotated samples. On this point, in our work, we demonstrate that in-context learning is highly effective in the data-starved regime (32 samples or fewer). There may be opportunities to combine different methods to leverage their complementary benefits, in particular when targeting less data-constrained data regimes (e.g., hundreds of samples).

**Extending the visual and text interface.** Natural language is a powerful and versatile input/output interface to provide descriptions of visual tasks to the model and generate outputs or estimate conditional likelihoods over possible outputs. However, it may be a cumbersome interface for tasks that involve conditioning on or predicting more structured outputs such as bounding boxes (or their temporal and spatio-temporal counterparts); as well as making spatially (or temporally and spatio-temporally) dense predictions. Furthermore, some vision tasks, such as predicting optical flow, involve predicting in continuous space, which is not something our model is designed to handle out of the box. Finally, one may consider additional modalities besides vision that may be complementary, such as audio. All of these directions have the potential to extend the range of tasks that our models can handle; and even improve performance on the ones we focus on, thanks to synergies between the corresponding abilities.

**Scaling laws for vision-language models.** In this work, we scale Flamingo models up to 80B parameters and provide some initial insights on their scaling behaviour across evaluation benchmarks, summarized in Figure 2. In the language space, an important line of work has focused on establishing scaling laws for language models [42, 53]. In the vision domain, Zhai et al. [145] take a step in this direction. Similar efforts have yet to be made for vision-language models, including contrastive models, as well as visual language models such as the ones we propose. While language modeling scaling law research has focused on perplexity as the golden metric, we speculate that it may be more directly useful for our purposes to establish such trends in terms of aggregate downstream evaluation task performance.

### D.2 Benefits, risks and mitigation strategies

#### D.2.1 Benefits

**Accessibility.** A system like Flamingo offers a number of potential societal benefits, some of which we will discuss in this section. Broadly, the fact that Flamingo is capable of task generalisation makes it suitable for use cases that have not been the focus of vision research historically. Typical vision systems are trained to solve a particular problem by training on large databases of manually annotated task-specific examples, making them poorly suited for applications outside of the narrow use cases for which they were deliberately trained. On the other hand, Flamingo is trained in a minimally constrained setting, endowing it with strong few-shot task induction capabilities. As we've shown in our qualitative examples (Appendix C), Flamingo can also be used through a "chat"-like interface for open-ended dialogue. Such capabilities could enable non-expert end users to apply models like Flamingo even to low-resource problems for which little to no task-specific training data has been collected, and where queries might be posed in a variety of formats and writing styles. In this direction, we have shown that *Flamingo* achieves strong performance on the VizWiz challenge[1], which promotes visual recognition technologies to assist visually impaired people. A dialogue interface could also promote better understanding and interpretability of visual language models. It could help highlight issues with bias, fairness, and toxicity the model may pick up on from the training data. Overall, we believe that Flamingo represents an important step towards making state-of-the-art visual recognition technology more broadly accessible and useful for many diverse applications.

**Model recycling.** From a modeling perspective, although Flamingo is computationally expensive to train, it importantly leverages pretrained frozen language models and visual encoders. We demonstrated that new modalities can be introduced into frozen models, thereby avoiding expensive retraining. As such models continue to grow in size and computational demands, "recycling" them will become increasingly important from an environmental perspective (as well as a practical one), as described in Larochelle [55] and explored in Strubell et al. [105] for language models. We hope such results may inspire further research into how existing models can be repurposed efficiently rather than trained from scratch.

#### D.2.2 Risks and mitigation strategies

This section provides some early investigations of the potential risks of models like Flamingo. This study is preliminary and we foresee that further research efforts should be undertaken to better assess those risks. We also discuss potential mitigation strategies towards safely deploying these models. Note that as explained in our Model Card [77] in Appendix E, this model was developed for research purposes only and should not be used in specific applications before proper risk analyses are conducted and mitigation strategies are explored.

**By construction, *Flamingo* inherits the risks of Large LMs.** Recall that a large part of our model is obtained by freezing the weights of an existing language model [42]. In particular, if provided with no images *Flamingo* falls back to language model behavior. As such *Flamingo* is exposed to the same risks of large language models: it can output potentially offensive language, propagate social biases and stereotypes, as well as leaking private information [126]. In particular, we refer to the analysis presented in the Chinchilla paper (Hoffmann et al. [42], Section 4.2.7) in terms of gender bias on the Winogender dataset [93] which demonstrate that even though this model is less biased towards gender than previous models [86], gender biases are still present. In terms of unprompted toxicity, we also refer to the analysis from Chinchilla [42] which highlights that overall the propensity of the model to produce toxic outputs when not prompted to do so is rather low, as measured by computing the *PerspectiveAPI* toxicity score on 25,000 samples. Weidinger et al. [126] detail possible long-term mitigation strategies for these risks. They include social or public policy interventions, such as the creation of regulatory frameworks and guidelines; careful product design, for instance relating to user interface decisions; and research at the intersection between AI Ethics and NLP, such as building better benchmarks and improving mitigation strategies. In the short term, effective approaches include relying on prompting to mitigate any biases and harmful outputs [86]. Next, we explore the additional risks incurred by Flamingo's additional visual input capabilities.

---

[1]https://vizwiz.org/

| | CIDEr difference | | CIDER |
| | female - male = $\Delta$ | darker - lighter = $\Delta$ | overall |
|---|---|---|---|
| AoANet [46] | - | +0.0019 | 1.198 |
| Oscar [61] | - | +0.0030 | 1.278 |
| *Flamingo*, 0 shot | $0.899 - 0.870 = +0.029$ $(p = 0.52)$ | $0.955 - 0.864 = +0.091$ $(p = 0.25)$ | 0.843 |
| *Flamingo*, 32 shots | $1.172 - 1.142 = +0.030$ $(p = 0.54)$ | $1.128 - 1.152 = -0.025$ $(p = 0.76)$ | 1.138 |

Table 12: **Bias evaluation of *Flamingo* for COCO captioning.** We report results on the COCO dataset splits over gender and skin tone provided by Zhao et al. [147].

**Gender and racial biases when prompted with images.** Previous work has studied biases that exist in captioning systems [37, 147]. Such modeling biases can result in real-world harms if deployed without care. For AI systems to be useful to society as a whole, their performance should not depend on the perceived skin tone or gender of the subjects – they should work equally well for all populations. However, current automatic vision system performance has been reported to vary with race, gender or when applied across different demographics and geographic regions [12, 21, 97]. As a preliminary study assessing how Flamingo's performance varies between populations, we follow the study proposed in Zhao et al. [147] and report how the captioning performance of our model varies on COCO as a function of gender and race. Note that we use a different evaluation protocol from the one proposed by Zhao et al. [147]; in that work, they measure results across 5 pretrained models and compute confidence intervals across aggregated per-model scores. Here, we have just one copy of our model (due to its high training cost), and we instead perform statistical tests on the per-sample CIDEr scores across the splits from Zhao et al. [147]. We report the results in Table 12.

Overall, when comparing the CIDEr scores aggregated among images labeled as *female* versus *male*, as well as when comparing *darker skin* versus *lighter skin*, we find there are no statistically significant differences in the per-sample CIDEr scores. To compare the two sets of samples, we use a two-tailed $t$-test with unequal variance, and among the four comparisons considered, the lowest $p$-value we find is $p = 0.25$, well above typical statistical significance thresholds (e.g. a common rejection threshold might be $p < \alpha = 0.05$). This implies that the differences in scores are indistinguishable from random variation under the null hypothesis that the mean scores are equal. We note that a failure to reject the null hypothesis and demonstrate a significant difference does not imply that there are no significant differences; it is possible that a difference exists that could be demonstrated with larger sample sizes, for example. However, these preliminary results are nonetheless encouraging.

**Toxicity when prompted with images.** We also evaluate the toxicity of *Flamingo* using the *Perspective API*[2] to evaluate the toxicity of the model's generated captions when prompted with images from the COCO test set. We observe that some captions are labelled as potentially toxic by the classifier; however, when examining them manually, we do not observe any clear toxicity – output captions are appropriate for the images provided. Overall, based on our own experiences interacting with the system throughout the course of the project, we have not observed toxic outputs when given "safe-for-work" imagery. However this does not mean the model is incapable of producing toxic outputs, especially if probed with "not-safe-for-work" images and/or toxic text. A more thorough exploration and study would be needed if such a model were put in production.

**Applying Flamingo for mitigation strategies.** Thanks to its ability to rapidly adapt in low-resource settings, *Flamingo* could itself be applied in addressing some of the issues described above. For instance, following Thoppilan et al. [111], adequately conditioned or fine-tuned Flamingo models could be used for filtering purposes of toxic or harmful samples in the training data. In their work, they observe significant improvements relating to safety and quality when fine-tuning on the resulting data. Furthermore, during evaluation, such adapted models could be used to down-rank or exclude outputs that might be classified as offensive, promoting social biases and stereotypes or leaking private information, thus accelerating progress in this direction even for low-resource tasks. Our results on the HatefulMemes benchmark represent a promising step in this direction. Recent work in the language modeling space has also shown success in training an LM to play the role of a "red team" and generate test cases, so as to automatically find cases where another target LM behaves in a harmful way [81]. A similar approach could be derived for our setting. Enabling the model to support

---
[2]https://perspectiveapi.com/

outputs with reference to particular locations within the visual inputs, or to external verified quotes is also an interesting direction [72, 111]. Finally, in Figure 11, we provide qualitative examples demonstrating that Flamingo can explain its own outputs, suggesting avenues to explainability and interpretability using the model's text interface.

# E *Flamingo* Model Card

We present a model card for Flamingo in Table 13, following the framework presented by Mitchell et al. [77].

| Model Details | |
|---|---|
| Model Date | March 2022 |
| Model Type | Transformer-based autoregressive language model, conditioned on visual features from a convnet-based encoder. Additional transformer-based cross-attention layers incorporate vision features into the language model's text predictions. (See Section 2 for details.) |

| Intended Uses | |
|---|---|
| Primary Intended Uses | The primary use is research on visual language models (VLM), including: research on VLM applications like classification, captioning or visual question answering, understanding how strong VLMs can contribute to AGI, advancing fairness and safety research in the area of multimodal research, and understanding limitations of current large VLMs. |
| Out-of-Scope Uses | Uses of the model for visually conditioned language generation in harmful or deceitful settings. Broadly speaking, the model should not be used for downstream applications without further safety and fairness mitigations specific to each application. |

| Factors | |
|---|---|
| Card Prompts – Relevant Factor | Relevant factors include which language is used. Our model is trained on English data. Our model is designed for research. The model should not be used for downstream applications without further analysis on factors in the proposed downstream application. |
| Card Prompts – Evaluation Factors | *Flamingo* is based on Chinchilla (a large proportion of the weights of Chinchilla are used as this) and we refer to the analysis provided in [42, 86] for the language only component of this work. We refer to our study presented in Appendix D.2.2 for a toxicity analysis when the model is conditioned on an image. |

| Metrics | |
|---|---|

| Model Performance Measures | We principally focus on the model's ability to predict relevant language when given an image. For that we used a total of 18 different benchmarks described in Appendix B.1.4 spanning various vision and language tasks such as classification (ImageNet, Kinetics700, HatefulMemes), image and video captioning (COCO, VATEX, Flickr30K, YouCook2, RareAct), visual question answering (OKVQA, VizWiz, TextVQA, VQAv2, MSRVTTQA, MSVDQA, iVQA, STAR, NextQA) and visual dialog (VisDiag). This was tested either in an open ended setting where *Flamingo* generate language and we compare the outputs with the ground truth or in a close ended setting where we directly score various outcomes using the likelihood of the model. |
|---|---|
| Decision thresholds | N/A |
| Approaches to Uncertainty and Variability | Due to the costs of training *Flamingo*, we cannot train it multiple times. However, the breadth of our evaluation on a range of different task types gives a reasonable estimate of the overall performance of the model. |

### Evaluation Data

| Datasets | See Table 6 for a detailed list. |
|---|---|
| Motivation | We chose our evaluation datasets to span an important range of vision and language tasks to correctly assess the ability of *Flamingo* to produce relevant text given an image. |
| Preprocessing | Input text is tokenized using a SentencePiece tokenizer with a vocabulary size of 32,000. Images are processed so that their mean and variance are 0 and 1 respectively. |

### Training Data

See [50], the Datasheet in Appendix F.1, Appendix F.2.1, Appendix F.2.2

### Quantitative Analyses

| Unitary Results | *Flamingo* sets a new state of the art in few-shot learning on a wide range of open-ended vision and language tasks. On the 16 tasks we consider, Flamingo also surpasses the fine-tuned state-of-art in 6 of the cases despite using orders of magnitude less task-specific training data. We refer to Section 3 for the full details of our quantitative study. |
|---|---|
| Intersectional Results | We did not investigate intersectional biases. |

### Ethical Considerations

| Data | The data is sourced from a variety of sources, some of it from web content. Sexually explicit content is filtered out, but the dataset does include racist, sexist or otherwise harmful content. |
|---|---|
| Human Life | The model is not intended to inform decisions about matters central to human life or flourishing. |

| | |
|---|---|
| Mitigations | Apart from removing sexual explicit content we did not filter out toxic content, following the rationale of Rae et al. [86]. More work is needed on mitigation approaches to toxic content and other types of risks associated with language models, such as those discussed in Weidinger et al. [126]. |
| Risks and Harms | The data is collected from the internet, and thus undoubtedly toxic and biased content is included in our training dataset. Furthermore, it is likely that personal information is also in the dataset that has been used to train our models. We defer to the more detailed discussion in Weidinger et al. [126]. |
| Use Cases | Especially fraught use cases include the generation of factually incorrect information with the intent of distributing it or using the model to generate racist, sexist or otherwise toxic text with harmful intent. Many more use cases that could cause harm exist. Such applications to malicious use are discussed in detail in Weidinger et al. [126]. |

Table 13: **Flamingo Model Card.** We follow the framework presented in Mitchell et al. [77].

## F   Datasheets

### F.1   M3W dataset

We follow the framework defined by Gebru et al. [30] and provide the datasheet for *M3W* in Table 14.

| **Motivation** | |
|---|---|
| For what purpose was the dataset created? Who created the dataset? Who funded the creation of the dataset? | The dataset was created for pre-training vision-language models and was created by researchers and engineers. |
| Any other comments? | None. |
| **Composition** | |
| What do the instances that comprise the dataset represent (e.g., documents, photos, people, countries)? | All instances of the dataset are documents from the web containing interleaved text and images. |
| How many instances are there in total (of each type, if appropriate)? | There are 43.3M instances (documents) in total, with a total of 185M images and 182 GB of text. |
| Does the dataset contain all possible instances or is it a sample (not necessarily random) of instances from a larger set? | The dataset is a sample from a larger set. |
| What data does each instance consist of? | Each instance is made up of a sequence of UTF-8 bytes encoding the document's text, as well as a sequence of integers indicating the positions of images in the text, and the images themselves in compressed format (see Section 2.4). |
| Is there a label or target associated with each instance? | No, there are no labels associated with each instance. |

| | |
|---|---|
| Is any information missing from individual instances? | No. |
| Are relationships between individual instances made explicit? | There are no relationships between the different instances in the dataset. |
| Are there recommended data splits? | We use random splits for the training and development sets. |
| Are there any errors, sources of noise, or redundancies in the dataset? | There is significant redundancy at the sub-document level. |
| Is the dataset self-contained, or does it link to or otherwise rely on external resources? | The dataset is self-contained. |
| Does the dataset contain data that might be considered confidential? | No. |
| Does the dataset contain data that, if viewed directly, might be offensive, insulting, threatening, or might otherwise cause anxiety? | The dataset likely contains some data that might be considered offensive, insulting or threatening, as such data is prevalent on the web. We do not try to filter out such content, with the exception of explicit content, which we identify using dedicated filter. |

### Collection Process

| | |
|---|---|
| How was the data associated with each instance acquired? | The data is available publicly on the web. |
| What mechanisms or procedures were used to collect the data? | The data was collected using a variety of software programs to extract and clean the raw text and images. |
| If the dataset is a sample from a larger set, what was the sampling strategy? | We randomly subsample documents. |
| Over what timeframe was the data collected? | The dataset was collected over a period of several months in 2021. We do not filter the sources based on creation date. |
| Were any ethical review processes conducted? | No. |

### Preprocessing/cleaning/labeling

| | |
|---|---|
| Was any preprocessing/Cleaning/Labeling of the data done (e.g., discretization or bucketing, tokenization, part-of-speech tagging, SIFT feature extraction, removal of instances, processing of missing values)? | Yes — the pre-processing details are discussed in Appendix A.3.1. |
| Is the software used to preprocess/clean/label the instances available? | No. |

### Uses

| | |
|---|---|
| Has the dataset been used for any tasks already? | Yes, we use the dataset for pre-training multimodal language and vision models. |
| Is there a repository that links to any or all papers or systems that use the dataset? | No, the dataset has only been used to train the models in this paper. |

| What (other) tasks could the dataset be used for? | We do not foresee other usages of the dataset at this stage. |
|---|---|
| Is there anything about the composition of the dataset or the way it was collected and preprocessed/-cleaned/labeled that might impact future uses? | The dataset is static and thus will become progressively more "stale". For example, it will not reflect new language and norms that evolve over time. However, due to the nature of the dataset it is relatively cheap to collect an up-to-date version. |
| Are there tasks for which the dataset should not be used? | The dataset described in this paper contains English language text almost exclusively and therefore should not be used for training models intended to have multilingual capabilities. |

### Distribution

| Will the dataset be distributed to third parties outside of the entity (e.g., company, institution, organization) on behalf of which the dataset was created? | No. |
|---|---|

Table 14: *M3W* **Datasheet**. We follow the framework as presented by Gebru et al. [30].

## F.2 Image and video text pair datasets

### F.2.1  Datasheet for LTIP

**Motivation**

| | |
|---|---|
| For what purpose was the dataset created? Who created the dataset? Who funded the creation of the dataset? | The dataset was created for pre-training vision-language models and was created by researchers and engineers. |
| Any other comments? | None. |

**Composition**

| | |
|---|---|
| What do the instances that comprise the dataset represent (e.g., documents, photos, people, countries)? | All instances of the dataset are image-text pairs. |
| How many instances are there in total (of each type, if appropriate)? | The dataset contains 312M image-text pairs. |
| Does the dataset contain all possible instances or is it a sample (not necessarily random) of instances from a larger set? | The dataset is a sample from a larger set. |
| What data does each instance consist of? | Each instance is made up of a sequence of UTF-8 bytes encoding the document's text, and an image in compressed format (see Appendix A.3.3). |
| Is there a label or target associated with each instance? | No, there are no labels associated with each instance. |
| Is any information missing from individual instances? | No. |
| Are relationships between individual instances made explicit? | There are no relationships between the different instances in the dataset. |
| Are there recommended data splits? | We use random splits for the training and development sets. |
| Are there any errors, sources of noise, or redundancies in the dataset? | The data is relatively high quality but there is a chance that some instances are repeated multiple times. |
| Is the dataset self-contained, or does it link to or otherwise rely on external resources? | The dataset is self-contained. |
| Does the dataset contain data that might be considered confidential? | No. |
| Does the dataset contain data that, if viewed directly, might be offensive, insulting, threatening, or might otherwise cause anxiety? | The websites that were used for this dataset were carefully selected to avoid such content. However given the scale of the data it is possible that some data could be considered offensive or insulting. |

**Collection Process**

| | |
|---|---|
| How was the data associated with each instance acquired? | The data is available publicly on the web. |
| What mechanisms or procedures were used to collect the data? | The data was collected using a variety of software programs to extract and clean the raw text and images. |

| If the dataset is a sample from a larger set, what was the sampling strategy? | N.A. |
|---|---|
| Over what timeframe was the data collected? | The dataset was collected over a period of several months in 2021. We do not filter the sources based on creation date. |
| Were any ethical review processes conducted? | No. |

### Preprocessing/cleaning/labeling

| Was any preprocessing/Cleaning/Labeling of the data done (e.g., discretization or bucketing, tokenization, part-of-speech tagging, SIFT feature extraction, removal of instances, processing of missing values)? | Some automatic text formatting was applied to remove from the captions dates and locations that were not relevant to the training objective. |
|---|---|
| Is the software used to preprocess/clean/label the instances available? | No. |

### Uses

| Has the dataset been used for any tasks already? | Yes, we use the dataset for pre-training multimodal language and vision models. |
|---|---|
| Is there a repository that links to any or all papers or systems that use the dataset? | No, the dataset has only been used to train the models in this paper. |
| What (other) tasks could the dataset be used for? | We do not foresee other usages of the dataset at this stage. |
| Is there anything about the composition of the dataset or the way it was collected and preprocessed/cleaned/labeled that might impact future uses? | The dataset is static and thus will become progressively more "stale". For example, it will not reflect new language and norms that evolve over time. However, due to the nature of the dataset it is relatively cheap to collect an up-to-date version. |
| Are there tasks for which the dataset should not be used? | The dataset described in this paper contains English language text almost exclusively and therefore should not be used for training models intended to have multilingual capabilities. |

### Distribution

| Will the dataset be distributed to third parties outside of the entity (e.g., company, institution, organization) on behalf of which the dataset was created? | No. |
|---|---|

Table 15: **LTIP Datasheet**. We follow the framework as presented by Gebru et al. [30].

### F.2.2   Datasheet for VTP

<table>
<tr><td colspan="2" align="center">**Motivation**</td></tr>
<tr><td>For what purpose was the dataset created? Who created the dataset? Who funded the creation of the dataset?</td><td>The dataset was created for pre-training vision-language models and was created by researchers and engineers.</td></tr>
<tr><td>Any other comments?</td><td>None.</td></tr>
<tr><td colspan="2" align="center">**Composition**</td></tr>
<tr><td>What do the instances that comprise the dataset represent (e.g., documents, photos, people, countries)?</td><td>All instances of the dataset are video-text pairs.</td></tr>
<tr><td>How many instances are there in total (of each type, if appropriate)?</td><td>The dataset contains 27M video-text pairs.</td></tr>
<tr><td>Does the dataset contain all possible instances or is it a sample (not necessarily random) of instances from a larger set?</td><td>The dataset is a sample from a larger set.</td></tr>
<tr><td>What data does each instance consist of?</td><td>Each instance is made up of a sequence of UTF-8 bytes encoding the document's text, and a video in compressed format (see Appendix A.3.3).</td></tr>
<tr><td>Is there a label or target associated with each instance?</td><td>No, there are no labels associated with each instance.</td></tr>
<tr><td>Is any information missing from individual instances?</td><td>No.</td></tr>
<tr><td>Are relationships between individual instances made explicit?</td><td>There are no relationships between the different instances in the dataset.</td></tr>
<tr><td>Are there recommended data splits?</td><td>We use random splits for the training and development sets.</td></tr>
<tr><td>Are there any errors, sources of noise, or redundancies in the dataset?</td><td>The data is relatively high quality but there is a chance that some instances are repeated multiple times.</td></tr>
<tr><td>Is the dataset self-contained, or does it link to or otherwise rely on external resources?</td><td>The dataset is self-contained.</td></tr>
<tr><td>Does the dataset contain data that might be considered confidential?</td><td>No.</td></tr>
<tr><td>Does the dataset contain data that, if viewed directly, might be offensive, insulting, threatening, or might otherwise cause anxiety?</td><td>The websites that were used for this dataset were carefully selected to avoid such content. However given the scale of the data it is possible that some data could be considered offensive or insulting.</td></tr>
<tr><td colspan="2" align="center">**Collection Process**</td></tr>
<tr><td>How was the data associated with each instance acquired?</td><td>The data is available publicly on the web.</td></tr>
<tr><td>What mechanisms or procedures were used to collect the data?</td><td>The data was collected using a variety of software programs to extract and clean the raw text and videos.</td></tr>
</table>

| | |
|---|---|
| If the dataset is a sample from a larger set, what was the sampling strategy? | N.A. |
| Over what timeframe was the data collected? | The dataset was collected over a period of several months in 2021. We do not filter the sources based on creation date. |
| Were any ethical review processes conducted? | No. |

### Preprocessing/cleaning/labeling

| | |
|---|---|
| Was any preprocessing/Cleaning/Labeling of the data done (e.g., discretization or bucketing, tokenization, part-of-speech tagging, SIFT feature extraction, removal of instances, processing of missing values)? | Some automatic text formatting was applied to remove from the captions dates and locations that were not relevant to the training objective. |
| Is the software used to preprocess/clean/label the instances available? | No. |

### Uses

| | |
|---|---|
| Has the dataset been used for any tasks already? | Yes, we use the dataset for pre-training multimodal language and vision models. |
| Is there a repository that links to any or all papers or systems that use the dataset? | No, the dataset has only been used to train the models in this paper. |
| What (other) tasks could the dataset be used for? | We do not foresee other usages of the dataset at this stage. |
| Is there anything about the composition of the dataset or the way it was collected and preprocessed/cleaned/labeled that might impact future uses? | The dataset is static and thus will become progressively more "stale". For example, it will not reflect new language and norms that evolve over time. However, due to the nature of the dataset it is relatively cheap to collect an up-to-date version. |
| Are there tasks for which the dataset should not be used? | The dataset described in this paper contains English language text almost exclusively and therefore should not be used for training models intended to have multilingual capabilities. |

### Distribution

| | |
|---|---|
| Will the dataset be distributed to third parties outside of the entity (e.g., company, institution, organization) on behalf of which the dataset was created? | No. |

Table 16: **VTP Datasheet**. We follow the framework as presented by Gebru et al. [30].

## G   Credit for visual content

- Figure 1:
  - Row 1: All images are provided under license by Unsplash.
  - Row 2: All images are under the public domain.

- – Row 3: First two images are provided under license by Unsplash.
- – Row 5: Available from DALL·E 2 [89].
- – Row 6: First two are provided under license by Unsplash, the third one is provided by Wikimedia Commons, licensed under CC BY-ND 2.0.
- – Row 7: The images are provided by Wikimedia Commons, licensed under CC BY-ND 2.0.
- – Row 8: The images are provided by Wikimedia Commons, licensed under CC BY-ND 2.0.
- – Row 9: This video is from YFCC100M, licensed under CC BY-ND 2.0.
- – Dialogue 1: Available from DALL·E 2 [89].
- – Dialogue 2: The first icon is provided under license by Flaticon, the second image is provided under license by Unsplash, the third one is provided under license by Sketchfab.
- – Dialogue 3: Available from CLIP [85].
- – Dialogue 4: Chicago and Tokyo pictures obtained from Unsplash.
- Model Figures 3, 7, 9 and 8: All images are provided under license by Unsplash.
- Qualitative Figures 10, 11, 12, and 13: All visuals are sourced from various sources including the COCO dataset, Wikimedia Commons, licensed under CC BY-ND 2.0 or available from DALL·E 2 [89].