# OpenReview forum: "Flamingo: a Visual Language Model for Few-Shot Learning"
_NeurIPS.cc/2022/Conference — NeurIPS 2022 Accept_

### Official Review · Reviewer_T9XK · 2022-07-12

**Rating:** 8
**Confidence:** 4
**Soundness:** 4 excellent
**Presentation:** 4 excellent
**Contribution:** 4 excellent

**Summary:**

The Flamingo model is a general few-shot solution for multimodal tasks. As input, the model can accept a multimodal prompt. The prompt contains a few examples of the desired task, with text, images, or videos as tokens. The model determines the task dynamics from the prompt and infers the desired task output. This model relies on three key ideas: (i) A frozen Pretrained Language Model (PLM) for training extensive knowledge, (ii) Reducing visual input to a fixed number of visual tokens with Perceiver, and (iii) Training on a web-based noisy image-text dataset (ALIGN) and a cleaner dataset (LTIP).

**Questions:**

Given the limited access to large-scale training data (e.g., ALIGN), how do the authors see future research on this topic?  Are the authors able to provide results from public data, such as LAION?


**Limitations:**

Yes

**Strengths And Weaknesses:**

Strengths:
The paper is generally of high quality. Various tasks are studied in depth. Compared to previous few-shot methods, the performance is significantly better. Additionally, the work includes an extensive ablation study of different design choices, including pertaining, training data mixtures, mixing different tasks, and architecture choices. Moreover, the method is efficient and uses frozen PLM and vision encoders. The only trainable parts are intermediate modules (e.g., Perceiver and gated XATTN-Dense attention).

Notably, I find few-shot supervision, and steering of large-scale PLM to solve different tasks to be human-like and an important research direction.


Weaknesses:



(W1) Data access is a major concern. As far as I know, the ALIGN dataset is not publicly available, so future research in this area will be limited. Data is a key factor for training, as indicated in the paper.

(W2) A similar work, "Multimodal Few-Shot Learning with Frozen Language Models," has many related ideas (e.g., frozen PLM, few-shot multimodal prompt solution). In addition, the study introduces a relevant benchmark, Open-Ended miniImageNet 2-Way Tasks.
While the paper does cite this work, the paper didn't elaborate enough on the fundamental differences between the studies. Is it only using new inputs such as videos, as mentioned on L275?

(W3) This is not a weakness, but a matter of taste. I found it a bit exhausting to go back and forth between the main paper and the appendix, especially with the appendix's size.  A concise but formal description of the method in the main paper (e.g., GATED XATTN-DENSE) would be greatly appreciated.

---

> ### Author Response · Authors · 2022-08-02
> **Response to reviewer T9XK**
>
> We thank the reviewer for their thoughtful feedback. Please see detailed responses below.
>
> > (W1) Data access is a major concern. As far as I know, the ALIGN dataset is not publicly available, so future research in this area will be limited. Data is a key factor for training, as indicated in the paper.
> > [...]
> > Given the limited access to large-scale training data (e.g., ALIGN), how do the authors see future research on this topic? Are the authors able to provide results from public data, such as LAION?
>
> We acknowledge that similar to prior work in this area (such as Instagram-1B-Targeted [1], CLIP [2], ALIGN [3]), the use of internal datasets poses some challenges for external reproduction of our work. It is nevertheless our conviction that sharing our findings publicly is beneficial to the community, as we hope that our results motivate public large-scale data collection endeavours like LAION.
>
> We are also interested in exploring the publicly available LAION dataset and training some models as this would provide valuable data points for people that would like to reproduce our work. If accepted, we will do our best to incorporate this ablation into the camera-ready version (we could not get those results in time for the rebuttal).
>
> > (W2) A similar work, "Multimodal Few-Shot Learning with Frozen Language Models," has many related ideas (e.g., frozen PLM, few-shot multimodal prompt solution). In addition, the study introduces a relevant benchmark, Open- Ended miniImageNet 2-Way Tasks. While the paper does cite this work, the paper didn't elaborate enough on the fundamental differences between the studies. Is it only using new inputs such as videos, as mentioned on L275?
>
> Substantial differences from the referenced work include our use of interleaved M3W training data to learn to condition outputs on interleaved sequences of visuals and text, as well as a different mechanism (GATED-XATTN-DENSE & Perceiver Resampler) for conditioning the LM on visual inputs. In particular, Frozen uses the pre-trained LM as-is, while we inject the GATED-XATTN-DENSE layers into the LM and train them from scratch, leading to higher capacity and more effective conditioning. Beyond these architectural differences, the support for video via the Perceiver Resampler, our use of pretrained contrastive models for visual representations, and the larger scale of our models and data also differentiate our approach. These design choices (several of which we ablate in Sec 3.3, Appendix B.3) contribute to Flamingo's greatly improved performance on downstream tasks compared to the Frozen paper (e.g., on VQAv2 Frozen gets 25.9% top-1 in zero-shot, while Flamingo-3B gets 49.2% and Flamingo-80B gets 56.3%).
>
> > (W3) This is not a weakness, but a matter of taste. I found it a bit exhausting to go back and forth between the main paper and the appendix, especially with the appendix's size. A concise but formal description of the method in the main paper (e.g., GATED XATTN-DENSE) would be greatly appreciated.
>
> In sec. 2.1.2 and 2.1.3 we did try to provide as detailed a description of the model as possible given space constraints, but unfortunately still had to leave some details to the appendix. We acknowledge this is not ideal. In our updated revision we have improved the Appendix reference to refer directly to the GATED-XATTN-DENSE subsection of the Appendix (A.1.2, which includes the Figure), and combined the main text and appendix into a single document with hyperlinked cross-references, which we hope makes this slightly more convenient for readers. In the camera-ready version we will use some of our extra content page to expand the main paper discussion on this and other architectural components.
>
> [1] Yalniz et al. 2019 [Billion-scale semi-supervised learning for image classification](https://arxiv.org/pdf/1905.00546v1.pdf).
>
> [2] Radford et al., 2021. [Learning Transferable Visual Models From Natural Language Supervision](https://arxiv.org/abs/2103.00020).
>
> [3] Jia et al., 2021. [Scaling Up Visual and Vision-Language Representation Learning With Noisy Text Supervision](https://arxiv.org/abs/2102.05918).

---

> > ### Comment · Reviewer_T9XK · 2022-08-10
> > **Thanks for the response**
> >
> > I believe the discussion on reproducibility and open sourcing is beyond the scope of this discussion, and it's best left for the community to address. I think that continuing to advance models based on data without access isn't the best way to impact the research community significantly. Pioneers in heavily computed fields are also responsible for devising suitable methods for evaluating and releasing public models. In this case, I am satisfied with releasing results for a model trained using LAION. In my opinion, This model weights should be released. As the authors promised to include it in the final revision, I retain my original score.

---

### Official Review · Reviewer_mm3o · 2022-07-15

**Rating:** 7
**Confidence:** 4
**Soundness:** 4 excellent
**Presentation:** 4 excellent
**Contribution:** 4 excellent

**Summary:**

This work introduced a new vision-language pretraining model, Flamingo, that aims at few-shot generalization on interleaved visual and textual data. The main architecture is based on Perceiver Resampler to connect a frozen language model and a visual encoder. In order to generate text conditioned on visual representations, the authors design a gated cross-attention module that takes in visual outputs within the pre-trained LM. The model is trained via multi-task joint training. A large-scale interleaved text and image dataset, M3W, is collected to enhance the few-shot capabilities. Experiments show that the proposed model outperforms prior state-of-the-art models on 16 tasks and surpasses the fine-tuned version on 6 tasks.

**Questions:**

- The model can only be used in uni-modal/multi-modal to text tasks, will and how the model can be generalized to a unified VL architecture, e.g. text to image, image to image?
- Can prompt engineering techniques that are used for GPT-3 (or similar models) be applied to Flamingo?

**Limitations:**

The authors provide sufficient analysis of the potential limitations of the proposed method, including massive architecture seeking, and legacies of large language models.
The requirements of huge amounts of TPUv4 hardware make the model hard to be reproduced by acamadic researchers.

**Strengths And Weaknesses:**

Strengths:
- The demonstrated few-shot capabilities on vision-language tasks are impressive and can refresh current VL performance on various multimodal benchmarks.
- The proposed architecture and gated cross-attention injected language models are technically sound and can be borrowed by future research on vision-language modeling.
- Ablation studies on the proposed architecture demonstrated the effectiveness of each module design, which also provides further guidance on future VL architecture design.
- The interleaved image-text dataset M3W is beneficial to VL community for future research on zero-shot learning.

Weaknesses:
- Similar to GPT-3 and other generative pretraining models, Flamingo requires a huge amount of model parameters (the smallest one is 1.4B and the performance drop is significant). How to distill the model to a smaller size may be worth discussing.
- Most dataset and tasks are perception level understanding, such as VQA. How is the reasoning capability of the model, for example on Visual commonsense Reasoning?

---

> ### Author Response · Authors · 2022-08-02
> **Response to reviewer mm3o (1/2)**
>
> We thank the reviewer for their thoughtful feedback. Please see detailed responses below.
>
> > Similar to GPT-3 and other generative pretraining models, Flamingo requires a huge amount of model parameters (the smallest one is 1.4B and the performance drop is significant). How to distill the model to a smaller size may be worth discussing.
>
> Distillation for large LMs is an active research area. "Gopher" [1] sec. G.2 shows this is difficult to achieve in such settings. Given our use of pretrained frozen LMs as a component of Flamingo, we expect it would be similarly difficult to achieve good distillation results in our setting. However, we certainly agree this is an important and interesting direction for future work.
>
> > Most dataset and tasks are perception level understanding, such as VQA. How is the reasoning capability of the model, for example on Visual Commonsense Reasoning?
>
> We do evaluate on a few datasets/tasks that arguably measure some reasoning capabilities, for example: HatefulMemes, NextQA, and STAR ("Situated Reasoning in Real-World Videos").
>
> Thanks for the suggestion to evaluate on Visual Commonsense Reasoning (VCR). VCR includes images of scenes with multiple people, and labeled bounding box locations for each person mentioned in the questions/answers, with references to these people in the question/answer texts, e.g. "Why is `[person4]` pointing at `[person1]`?". Flamingo has no built-in means of incorporating the box coordinate or person reference information, so evaluating Flamingo on this task is a bit tricky, and may require further research to address adequately.
>
> In the following table we present a set of *baseline* few-shot VCR results *without access to the locations of each person in the image*; i.e. just feeding in the unannotated VCR image and question/answer pairs, as in a typical VQA dataset. The only preprocessing applied to the question/answer texts was converting the `[personN]` tags to a more natural-language-like format: e.g., `[person4]` becomes `person 4`. These numbers can be compared to the 45.5% number reported in the VCR paper for a fine-tuned VQA model which also doesn't make use of the bounding box information (but is trained on the full VCR training split).
>
> | Flamingo VCR *Baseline*  | 4 shot | 8 shot | 16 shot | 32 shot |
> |---------------|--------|--------|---------|---------|
> | Flamingo 3B   | 42.4%  | 42.7%  | 42.8%   | 42.7%   |
> | Flamingo 9B   | 45.6%  | 45.7%  | 45.4%   | 45.7%   |
> | Flamingo      | 49.3%  | 49.6%  | 50.0%   | 50.5%   |
>
> We expect that these baseline results could be improved by incorporating the ignored information given in the box coordinates and person references. Conditioning our models on such localization information is an important direction for future work.
>
> > The model can only be used in uni-modal/multi-modal to text tasks, will and how the model can be generalized to a unified VL architecture, e.g. text to image, image to image?
>
> This is an interesting research direction for future work. Flamingo indeed focuses on visual recognition tasks which can be cast as vision-conditioned text generation problems, with the goal of exploiting pretrained LMs and designing our architecture to take advantage of the efficiency gains of unimodal text output. That said, Flamingo could be used for text-to-image retrieval tasks by comparing text likelihoods across a given set of possible images (as we use for classification of possible text responses, Appendix B.2.1, Table 7), but we have not performed comprehensive evaluations of this capability. This is an interesting direction for future work to make Flamingo-type models even more general.
>
> > Can prompt engineering techniques that are used for GPT-3 (or similar models) be applied to Flamingo?
>
> We have used prompt engineering successfully for visual dialogue (see Fig. 1 and Appendix Fig. 11; dialogue prompt in Appendix B.1.6). We've also noticed that altering the presented visual dialogue prompt affects the model's behavior and response style significantly. For the few-shot results, we tried to avoid prompt engineering as much as possible, as our goal was to rigorously measure the model's ability to generalize to new tasks from the given examples with minimal/no task-specific tuning. However, we do expect that further improvements on our reported zero/few-shot evaluation numbers would be possible with task-specific prompts. (We have explored using a handwritten task description for zero-shot COCO captioning with some success, but we observe that task-agnostic few-shot learning outperforms the zero-shot task description setup with <=4 in-context examples.)

---

> > ### Author Response · Authors · 2022-08-02
> > **Response to reviewer mm3o (2/2)**
> >
> > > The requirements of huge amounts of TPUv4 hardware make the model hard to be reproduced by academic researchers
> >
> > We acknowledge that the large-scale nature of our work poses some challenges for external reproduction by academic labs. However, recent efforts such as BLOOM from the BigScience project have used large-scale academic public access clusters (Jean-Zay in France) to successfully train GPT-3-size models. In addition, we do believe that such large-scale endeavours are worth publishing as they can open new research avenues to groups with less hardware capacity by providing ambitious targets for work focusing on improving data- and/or compute-efficiency of models and losses. This has been the case in the past with work such as fastText [2], which was shown to perform on par with deep learning classifiers in terms of accuracy, while being orders of magnitude faster for training and evaluation.
> >
> > [1] Rae et al., 2021. [Scaling Language Models: Methods, Analysis & Insights from Training Gopher](https://arxiv.org/abs/2112.11446).
> >
> > [2] Joulin et al. 2016. [Bag of Tricks for Efficient Text Classification](https://arxiv.org/abs/1607.01759).

---

### Official Review · Reviewer_agGw · 2022-07-16

**Rating:** 8
**Confidence:** 4
**Soundness:** 3 good
**Presentation:** 4 excellent
**Contribution:** 4 excellent

**Summary:**

This paper proposes to align a pretrained image encoder and a large pretrained generative text encoder to the same multimodal space, by training an additional perceiver resampler and gated cross-attention layers. The method is simple and effective. Experiments on various vision-language multimodal tasks show that the proposed model called Flamingo outperforms previous methods under the zero-shot and few-shot settings, and also achieves SOTA performances on several tasks under the finetuned setting.


**Questions:**

Will the trained models and pretraining datasets be released?

**Ethics Review Area:**

["I don’t know"]

**Limitations:**

Yes

**Strengths And Weaknesses:**

Strengths
1. The paper is quite well written and easy to follow.
2. The proposed method is simple and effective.
3. The proposed vision-language pretraining dataset with interleaved images and texts is novel, and would be valuable to the community.
4. Extensive experiments on various tasks and model sizes show the efficacy of the proposed model.

Weakness:
1. One of my main concerns is about the novelty. Aligning a pretrained image encoder and a pretrained text encoder to the same multimodal space has already been studied in previous works, though not at a scale as large as Flamingo. In particular, a very similar idea is proposed in [1], which aligns the CLIP image encoder and a pretrained BART model to the same model space via distillation methods with light additional projection heads. Similar to Flamingo, [1] also exhibits remarkable zero-shot performance on image caption and open-end VQA tasks. It would be better if this submission can clarify more on the connection and difference with [1] and other similar works.
2. The good performance of the proposed models depends heavily on the large model size and large pretraining dataset size. This makes the pretraining and inference of these models very expensive.
3. Some details about the datasets and experimental settings on downstream tasks are not very clear. E.g,
(1) it is not clear what are the sources for collecting the interleaved image-text dataset.
(2) Previous dual-stream vision-language models like CLIP are efficient for large-scale image-text retrieval because features can be precomputed. However, the image and texts are intertwined in the proposed Flamingo, and it is not clear how the proposed model can be used for retrieval tasks.

[1] Enabling Multimodal Generation on CLIP via Vision-Language Knowledge Distillation, ACL Findings 2022.

---

> ### Author Response · Authors · 2022-08-02
> **Response to reviewer agGw (1/2)**
>
> We thank the reviewer for their thoughtful feedback. Please see detailed responses below.
>
> > Aligning a pretrained image encoder and a pretrained text encoder to the same multimodal space has already been studied in previous works, though not at a scale as large as Flamingo. In particular, a very similar idea is proposed in [1], which aligns the CLIP image encoder and a pretrained BART model to the same model space via distillation methods with light additional projection heads.
>
> Thank you for bringing this work to our attention. In our revised manuscript we have referenced it in the Related Work discussion. This work is concurrent with our submission. While certainly relevant in its use of pretrained image and text representations for multimodal learning, we feel that Flamingo remains novel and differs from this work in several important respects. In particular, we demonstrate that Flamingo exhibits few-shot learning capabilities for the first time among such approaches. These few-shot learning capabilities enable Flamingo to handle new tasks that would likely be infeasible in the zero-shot regime, and are made possible by Flamingo's flexible cross-attention architecture, allowing it to be applied to "interleaved" multimodal data (arbitrary sequences of text and images/videos), used in both our M3W training dataset and our few-shot evaluations. Flamingo is also capable of handling video inputs.
>
> > The good performance of the proposed models depends heavily on the large model size and large pretraining dataset size. This makes the pretraining and inference of these models very expensive.
>
> We acknowledge that the proposed models are computationally expensive to train; improving efficiency is an important direction for future research. However, the training cost is amortized by the resulting models' ability to generalize to many tasks through few-shot in-context inference:  a single inference pass is relatively cheap compared with the cost of training a new model for each downstream task of interest. Furthermore, an important advantage of our setup is our use of novel architectural components like GATED-XATTN-DENSE which facilitate the ability to "recycle" pretrained vision and text representations (see our discussion on model "Recycling" in Appendix D.2). We learn only the "glue" between these pretrained models, thereby saving large additional retraining costs for these components. (Note that our ablations show that co-training the language model from scratch results in worse performance, while also being more computationally expensive.)
>
> > Some details about the datasets and experimental settings on downstream tasks are not very clear. E.g, (1) it is not clear what are the sources for collecting the interleaved image-text dataset.
>
> While we unfortunately can’t provide the list of webpages we started from to build M3W, as it is based on an internal database of pre-selected public content, we do provide all implementation details on the filtering stages that we employ to retain the highest quality content (see the Datasheet for M3W in Appendix F.1). We acknowledge that similar to prior work in this area (such as Instagram-1B-Targeted [2], CLIP [3], ALIGN [4]), this poses some challenges for external reproduction of our work. For this reason, we do not include our dataset as a contribution of our work. It is nevertheless our conviction that sharing our findings publicly is beneficial to the community, as we hope that our results will motivate a public large-scale data collection endeavour, following the one we have undertaken in the context of this project.
>
> > (2) Previous dual-stream vision- language models like CLIP are efficient for large-scale image-text retrieval because features can be precomputed. However, the image and texts are intertwined in the proposed Flamingo, and it is not clear how the proposed model can be used for retrieval tasks.
>
> Retrieval (and relatedly, classification) with Flamingo is possible, but unfortunately slow and likely worse than contrastive models for this purpose. Please see our discussion on limitations in classification performance in Appendix D.1 for more details on this. This is an interesting direction we hope to address in future research.
>
> > Will the trained models and pretraining datasets be released?
>
> We have no current plans to release these, but we believe that the amount of implementation detail provided in the submission will enable successful reproduction of our results.

---

> > ### Author Response · Authors · 2022-08-02
> > **Response to reviewer agGw (2/2)**
> >
> > [1] Dai et al., 2022. [Enabling Multimodal Generation on CLIP via Vision-Language Knowledge Distillation](https://arxiv.org/abs/2203.06386).
> >
> > [2] Yalniz et al. 2019. [Billion-scale semi-supervised learning for image classification](https://arxiv.org/pdf/1905.00546v1.pdf).
> >
> > [3] Radford et al., 2021. [Learning Transferable Visual Models From Natural Language Supervision](https://arxiv.org/abs/2103.00020).
> >
> > [4] Jia et al., 2021. [Scaling Up Visual and Vision-Language Representation Learning With Noisy Text Supervision](https://arxiv.org/abs/2102.05918).

---

> > > ### Comment · Reviewer_agGw · 2022-08-08
> > > **response**
> > >
> > > I'd like to thank the authors for answering my questions.  Overall, the authors largely addressed my concerns. While I do still share similar concerns as other reviewers (such as the sources for data collection, and the availability of model/data), I believe there are some worthwhile contributions in this paper.

---

### Official Review · Reviewer_ZdRp · 2022-07-17

**Rating:** 8
**Confidence:** 5
**Soundness:** 4 excellent
**Presentation:** 4 excellent
**Contribution:** 4 excellent

**Summary:**

This paper introduces an interesting way of combining large, pre-trained vision and language models for solving a wide range of multi-modal tasks. The proposed model, Flamingo, is able to ingest a sequence of text tokens interleaved with images and/or videos, and produces text as output. The authors further demonstrate the efficiency of this model on a wide range of visual-language benchmarks such as few-shot image captioning, reasoning and question-answering tasks where it outperforms many SOTA models.

**Questions:**

- The few-shot VQA performance of Flamingo is impressive. However, I have a feeling that the model needs to be guided with text prompts to answer certain aspects related to an image. Is it possible for the model to answer a certain question without building the context? For e.g. In Fig.1 re: Flamingo image, what would be model response if you directly ask $\textit{what is the difference between these three images?}$ without the initial leading questions. Similarly, in Fig. 11 re: street sign in NYC, what is the response if you directly ask $\textit{Which park is nearby?}$

- What is the model performance if you don't use pre-trained vision and language models i.e. training them from scratch with Perceiver Resampler etc.? Does the proposed framework not help in extracting useful low-level features from images and text?

- Is it possible to visualise how textual embeddings map (or attend) to visual embeddings? I reckon, given the task (query prompt) this attention will change for the same image.

- How does the pre-trained visual or language model affects (or limits) the performance of Flamingo? You have mentioned this in the limitations section but have you tried ablating those components? or do you have a failure example?

- I am surprised that spatial grid positional encodings did not help while flattening 2D spatial features to a 1D sequence. In general, this should result in the loss of valuable spatial information. Can authors provide some insights on this?

- What's the reason behind choosing NFNet for visual encoder? Did you try transformers (e.g. ViT) as a vision encoder where you can directly utilise tokens as inputs to Perceiver Resampler?

- What does datasets $D_{m}$ refer to in Eq. 2? I thought the model is trained on only M3W dataset, constituting data from multiple documents. After some sifting through Appendix A, I found the relevant details in Sec. A.3.3. I think it would be a good idea to mention it in the Sec. 2.4 of main paper before introducing the equation.

- For training, do you need the data to be arranged in certain task-specific order for it to generalize to the tasks in few-shot manner during evaluation?

- Out of curiosity, what’s the idea behind naming this model Flamingo?

**Limitations:**

Authors have adequately highlighted the limitations and societal impacts of this work.

**Strengths And Weaknesses:**

**Strengths:**
- The architecture for combining pre-trained visual and language models, while preserving their knowledge, is novel and would be beneficial for the community in adapting large pre-trained models effectively.
- The paper empirically justifies almost all the aspects of design choices via adequate evaluations.
- The proposed method demonstrates impressive performance on few-shot image captioning, reasoning and question-answering tasks.
- The paper is very detailed, well-written and easy to follow..

**Weaknesses:**
I do not have major concerns with the paper but a suggestion to improve the readability of the paper. I have put my questions in the `questions` section below.
- It’s hard to navigate the appendix of the paper. Could you please mention the exact sections of appendix for specific details while referencing it in the main paper? For e.g. see Appendix A.1.3 instead of just Appendix A.

---

> ### Author Response · Authors · 2022-08-02
> **Response to reviewer ZdRp (1/2)**
>
> We thank the reviewer for their thoughtful feedback. Please see detailed responses below.
>
> > It’s hard to navigate the appendix of the paper. Could you please mention the exact sections of appendix for specific details while referencing it in the main paper? For e.g. see Appendix A.1.3 instead of just Appendix A.
>
> Sorry for the navigation difficulties. In the revised manuscript, we have updated our appendix references to point to more specific subsections where possible, and also combined the main text and appendix into a single document with hyperlinked cross-references.
>
> > The few-shot VQA performance of Flamingo is impressive. However, I have a feeling that the model needs to be guided with text prompts to answer certain aspects related to an image.
>
> For all quantitative VQA evaluations, the few-shot prompt is built by selecting random examples from the training set (image, question, answer), without any attempt to select ones that improve performance on target examples. The only additions to the training set texts are the use of "Question:" before the question and "Answer:" before the answer, as described in Appendix B.1.5.
>
> However, a prompt can indeed influence the models' behaviour -- in particular, note that all of our qualitative dialogue examples are conditioned on the same hand-crafted prompt containing a few images and example responses from Flamingo. This prompt, provided in Appendix B.1.6, dictates the style of responses from Flamingo in the presented dialogue examples. This is a conventional practice to adjust the behavior of large LMs, see GPT-3 [1] and Gopher [2] (in particular, Gopher uses a prompt to guide dialogue as we do for our Flamingo dialogue examples).
>
> > Is it possible for the model to answer a certain question without building the context? For e.g. In Fig.1 re: Flamingo image, what would be model response if you directly ask  "what is the difference between these three images?" without the initial leading questions.
>
> Without the initial "What is similar" question, when we ask *What is the difference between these three images?*, Flamingo responds: *The first one is a real flamingo, the second one is a low-poly flamingo, and the third one is a 3D model of a flamingo.* It seems to have flipped its descriptions of the first and second images, so this is incorrect. However, the structure of the response matches the original answer without the leading question.
>
> > Similarly, in Fig. 11 re: street sign in NYC, what is the response if you directly ask "Which park is nearby?"
>
> In this case the model responds "The closest park is Madison Square Park." (This is also incorrect as that's only the second closest park, so it is indeed plausible that the leading questions were helping the model to provide a more correct answer.)
>
> > What is the model performance if you don't use pre-trained vision and language models i.e. training them from scratch with Perceiver Resampler etc.? Does the proposed framework not help in extracting useful low-level features from images and text?
>
> Table 3 (main ablations table) includes the results of ablating the pretrained LM. Table 10 (additional ablations in Appendix B.3) includes the results of ablating the pretrained vision module. The numbers are substantially worse than the pretrained+frozen baselines; however, the results are certainly nontrivial, indicating that VLM training is able to train the vision encoder or the LM reasonably well, but of course less efficiently than using a pretrained model. Vision pretraining with a contrastive objective is quite data-efficient due to the use of large batches and negatives within batches. LLM pretraining is also efficient due to the use of text-only corpora.
>
> > Is it possible to visualise how textual embeddings map (or attend) to visual embeddings? I reckon, given the task (query prompt) this attention will change for the same image.
>
> It's likely possible but tricky due to the use of the Perceiver Resampler, which attends over the full spatial grid. The cross-attention added to the LM stack does not directly attend to the spatial grid, but to the non-spatial Perceiver Resampler outputs, making it more indirect to track the attended spatial locations. We leave this interesting direction for future work.

---

> > ### Author Response · Authors · 2022-08-02
> > **Response to reviewer ZdRp (2/2)**
> >
> > > How does the pre-trained visual or language model affects (or limits) the performance of Flamingo? You have mentioned this in the limitations section but have you tried ablating those components? or do you have a failure example?
> >
> > We certainly don't have a complete solution to the issues with LMs. We've observed that fine-tuning for VQA (for example) alleviates some LM bias; e.g. finetuning improves performance on counting-based questions substantially. One could also finetune or prompt Flamingo on tasks with responses like "I don't know" or "Trick question!" to alleviate hallucinations; however, we haven't explored this in depth.
> >
> > > I am surprised that spatial grid positional encodings did not help while attening 2D spatial features to a 1D sequence. In general, this should result in the loss of valuable spatial information. Can authors provide some insights on this?
> >
> > Note that in a convnet such as the NFNet we use, the padding in each layer, in conjunction with a wide receptive field at the end of the network, allows the model to infer some spatial information. This likely explains why we observed that introducing explicit position information doesn't give a further boost. In prior work, [3] has also shown that convnets implicitly encode spatial information channel-wise -- we have added this reference to our manuscript. We will add this reference to the camera-ready version of the paper to help clarify this.
> >
> > > What's the reason behind choosing NFNet for visual encoder? Did you try transformers (e.g. ViT) as a vision encoder where you can directly utilise tokens as inputs to Perceiver Resampler?
> >
> > Note that the outputs of NFNet and ViT are of essentially similar flavors: NFNet outputs a HxWxD 2D spatial grid of features, while ViT outputs can be viewed as (HW)xD features, where H and W are determined by the patching in the first ViT layer (which can be viewed as a convolution). The similarity may be more apparent once we flatten NFNet's HxWxD spatial grid to HWxD for input into the PerceiverResampler, matching the "raw" outputs of ViT. That said, NFNet proved to be faster on our hardware, and we ran into stability issues with ViT during the contrastive pretraining phase. We did experiment with CLIP ViT as an encoder; these results can be found in our ablations Table 3 "CLIP ViT-L/14". It outperforms the smaller NFNet-F0, but performs substantially below our main result with NFNet-F6.
> >
> > > What does datasets refer to in Eq. 2? I thought the model is trained on only M3W dataset, constituting data from multiple documents. After some sifting through Appendix A, I found the relevant details in Sec. A.3.3. I think it would be a good idea to mention it in the Sec. 2.4 of main paper before introducing the equation.
> >
> > We train on not only M3W, but Image-Text Pairs (ITP), and Video-Text Pairs (VTP) as well, as described in Sec 2.4 (L140-141). We compute the gradients on each of these datasets separately, accumulating these gradients to compute a single VLM parameter update. The discussion of the ITP/VTP datasets in the main paper was somewhat terse due to space limitations, and we will make sure to better clarify our use of these in the camera-ready version with the additional content page.
> >
> > > For training, do you need the data to be arranged in certain task-specific order for it to generalize to the tasks in few-shot manner during evaluation?
> >
> > If the question is about the order in which we provide the different datasets at training time (M3W, ITP and VTP), the answer is no as we accumulate gradients over all tasks. It's effectively equivalent to building a single batch containing a fixed number of elements from each task at each training step.
> >
> > If the question is more about the order of images in the interleaved samples from M3W, then no specific effort is made to make it better for downstream tasks: we simply order the images and the text as they appear in the webpage (see Appendix A.3.2 for details).
> >
> > > Out of curiosity, what’s the idea behind naming this model Flamingo?
> >
> > Flamingos change color depending on what they eat; our Flamingo model changes its behavior depending on the few-shot samples it "eats" :)
> >
> > [1] Brown et al., 2020. [Language Models are Few-Shot Learners](https://arxiv.org/abs/2005.14165).
> >
> > [2] Rae et al., 2021. [Scaling Language Models: Methods, Analysis & Insights from Training Gopher](https://arxiv.org/abs/2112.11446).
> >
> > [3] Islam et al., 2021. [Global pooling, more than meets the eye: Position information is encoded channel-wise in CNN](https://arxiv.org/abs/2108.07884).

---

### Official Review · Reviewer_KruY · 2022-07-24

**Rating:** 3
**Confidence:** 4
**Ethics Flag:** Yes
**Soundness:** 3 good
**Presentation:** 3 good
**Contribution:** 2 fair

**Summary:**

This paper introduces a new pre-trained vision-and-language model (VLM), Flamingo, built upon pre-trained vision and language models. The Flamingo model can take arbitrarily interleaved visual data and text as input and do open-ended text generation.
The Flamingo model shows promising results on multiple multimodal tasks in few-shot learning and can even outperform some SOTA models on some tasks.

**Questions:**

- Do the authors plan to release the model and/or the dataset used to train the model? If so, when?
- Do the authors plan to release an API like what OpenAI does to allow people to play with your model?
- How can one trust the results in the paper (not by faith) if no one can reproduce the results?

**Ethics Review Area:**

["Inadequate Data and Algorithm Evaluation", "Responsible Research Practice (e.g., IRB, documentation, research ethics)"]

**Limitations:**

- Missing details of the dataset introduced in the paper. No plan for the data and model release.
- Inadequate reproducibility.

See the weaknesses mentioned above for details.

**Strengths And Weaknesses:**

Strengths:
-
- This paper introduces a new pre-trained VLM built upon existing powerful pre-trained vision models and language models, and archives promising few-shot learning results on a wide array of vision-and-language tasks.
- Useful tricks are proposed for bridging pre-trained vision models and language models, including vision feature resampling, inserting cross-modal attention into language models, etc.
- A new dataset is also mentioned in the paper to pre-train the Flamingo model.
- The paper is well written in general and easy to understand.

Weaknesses:
-
- Taking arbitrarily interleaved visual data and text as input and generating text isn't new for VLM models; the CM3 [1] model does it already and even can do image generation in addition to text generation. Since CM3 is released months ago before the NeurIPS submission, the discussion about it in the paper isn't fair ("CM3 follows the paper's similar approach"). The existence of CM3 does not invalidate some of the main contributions of Flamingo, but it has to be fairly discussed, especially considering the similarities those two models have, e.g., the way to interleave and encode visual and text data in a sequence.
- However, lots of implementation and dataset details are missing (especially the dataset), so reproducing the results in the paper by a third party is nearly impossible.
- No data and model are attached, and no plan is mentioned in the paper. If the authors are not going to release the model and the data, I am not sure what the point is as no one else can reproduce the results (you have to trust the results reported in the paper blindly). Most researchers do not have the resources to re-train the model, and even with sufficient resources, a third party cannot reproduce the results without access to the data used in the paper.
NeurIPS advocates open source research and cares so much for reproducibility, so non-reproducible research is not acceptable.
In that case, this paper would be mainly a PR and flag-planting paper for a big institute/company, and does minimal good (and arguably hurt) the research community. I think such kinds of research have to convince people first about the validity and reproducibility of their results before being accepted to top conferences.
- Figure 1 is too good to be true. I know those results are most likely to be cherry-picked and supposed to be good. But without a detailed and fair analysis, they could be very misleading.

[1] CM3: A Causal Masked Multimodal Model of the Internet, Aghajanyan et al., 2022.

---

> ### Author Response · Authors · 2022-08-02
> **Response to reviewer KruY (1/2)**
>
> > No data and model are attached, and no plan is mentioned in the paper. If the authors are not going to release the model and the data, I am not sure what the point is as no one else can reproduce the results (you have to trust the results reported in the paper blindly). Most researchers do not have the resources to re-train the model, and even with sufficient resources, a third party cannot reproduce the results without access to the data used in the paper. NeurIPS advocates open source research and cares so much for reproducibility, so non-reproducible research is not acceptable. In that case, this paper would be mainly a PR and flag-planting paper for a big institute/company, and does minimal good (and arguably hurt) the research community. I think such kinds of research have to convince people first about the validity and reproducibility of their results before being accepted to top conferences.
>
> We respectfully disagree with the reviewer’s perspective. The reviewer has raised three main concerns: validity (trust), reproducibility and open sourcing, and the point of publishing work like this at all; we discuss each in turn.
>
> _Validity and trustworthiness_. The reviewer asks “How can one trust the results in the paper (not by faith) if no one can reproduce the results?” First of all, we do expect that the community will be able to reproduce our findings, otherwise we would not have submitted this scientific work for publication. On trustworthiness of the results, for many of our evaluation benchmarks (VQAv2, VizWiz, STAR, VisDial, TextVQA, etc.), test sets are hidden and we obtained our numbers by submitting to the official leaderboard. This considerably reduces the risk that there might be a mistake that would invalidate our findings.
>
> _Reproducibility and open sourcing_. There are numerous examples where work was rapidly and successfully reproduced by the community without published code, models or data: AlexNet [2] in Caffe [3]; GPT-3 [4], the best paper award winner at NeurIPS 2020, by Gopher [5] and OPT [6]; CLIP [7] by ALIGN [8] for zero-shot top-1 accuracy on ImageNet; AlphaGo [9] by ELF OpenGo [10]; etc. In addition, reproducibility and open-sourcing are different and one does not necessarily follow from the other: for instance, the most convincing proof of reproducibility of our work will be obtained when our architectural contributions are successfully adopted externally in different settings.
>
> _Why publish such work?_ More generally, we believe that the view that non-open-sourced work should not be published is dangerous and harmful to the field. Implementing such a policy would stall scientific progress by deterring researchers from large-scale undertakings like ours, where training data cannot be made publicly available and where systematic analysis and mitigation of possible harms should be done before committing to publishing models. We strongly believe that the paper provides enough technical details to be reproducible. We also present “an extensive ablation study” (RT9XK) comparing our design choices to other concurrent options. As stated by reviewer Rmm3o, this “provides further guidance on future VL architecture design” and saves precious exploration time to the community. In summary, we believe that our manuscript will be valuable for the scientific community.
>
> > However, lots of implementation and dataset details are missing (especially the dataset), so reproducing the results in the paper by a third party is nearly impossible.
>
> While we unfortunately can’t provide the list of webpages we started from to build M3W, as it is based on an internal database of pre-selected public content, we do provide all implementation details on the filtering stages that we employ to retain the highest quality content (see the Datasheet for M3W in Appendix F.1). We acknowledge that similar to prior work in this area (such as Instagram-1B-Targeted [11], CLIP dataset [7], ALIGN [8]), this poses some challenges for external reproduction of our work. For this reason, we do not include our dataset as a contribution of our work. It is nevertheless our conviction that sharing our findings publicly is beneficial to the community, as we hope that our results will motivate a public large-scale data collection endeavour, following the one we have undertaken in the context of this project.

---

> > ### Author Response · Authors · 2022-08-02
> > **Response to reviewer KruY (2/2)**
> >
> > > Figure 1 is too good to be true. I know those results are most likely to be cherry-picked and supposed to be good. But without a detailed and fair analysis, they could be very misleading.
> >
> > As is standard practice, these examples are selected to showcase the benefits of the approach. We have updated the figure caption in the revised manuscript to clarify this. For a fair and detailed analysis, we point the reviewer to the comprehensive quantitative evaluation that we perform across a range of diverse, widely-used and publicly available benchmarks, some of which are reserved until final evaluation to report unbiased few-shot learning performance. We also discuss limitations of our approach concisely in the main paper, and in detail in Appendix D.
> >
> > > Taking arbitrarily interleaved visual data and text as input and generating text isn't new for VLM models; the CM3 [1] model does it already and even can do image generation in addition to text generation. Since CM3 is released months ago before the NeurIPS submission, the discussion about it in the paper isn't fair ("CM3 follows the paper's similar approach"). The existence of CM3 does not invalidate some of the main contributions of Flamingo, but it has to be fairly discussed, especially considering the similarities those two models have, e.g., the way to interleave and encode visual and text data in a sequence.
> >
> > We did not intend for our phrasing to suggest that CM3 followed our approach. We will modify this description to avoid any ambiguity. We will also clarify the architectural differences between the two approaches.
> >
> >
> > [1] Aghajanyan et al., 2022. [CM3: A Causal Masked Multimodal Model of the Internet](https://arxiv.org/abs/2201.07520).
> >
> > [2] Krizhevsky et al., 2012. [ImageNet Classification with Deep Convolutional Neural Networks](https://papers.nips.cc/paper/2012/hash/c399862d3b9d6b76c8436e924a68c45b-Abstract.html).
> >
> > [3] Jia et al., 2014. [Caffe: Convolutional Architecture for Fast Feature Embedding](https://arxiv.org/abs/1408.5093).
> >
> > [4] Brown et al., 2020. [Language Models are Few-Shot Learners](https://arxiv.org/abs/2005.14165).
> >
> > [5] Rae et al., 2021. [Scaling Language Models: Methods, Analysis & Insights from Training Gopher](https://arxiv.org/abs/2112.11446).
> >
> > [6] Facebook AI, 2022. [Democratizing access to large-scale language models with OPT-175B](https://ai.facebook.com/blog/democratizing-access-to-large-scale-language-models-with-opt-175b/).
> >
> > [7] Radford et al., 2021. [Learning Transferable Visual Models From Natural Language Supervision](https://arxiv.org/abs/2103.00020).
> >
> > [8] Jia et al., 2021. [Scaling Up Visual and Vision-Language Representation Learning With Noisy Text Supervision](https://arxiv.org/abs/2102.05918).
> >
> > [9] Silver et al., 2017. [Mastering the game of Go without human knowledge](https://www.nature.com/articles/nature24270).
> >
> > [10] Tian et al., 2019. [ELF OpenGo: an analysis and open reimplementation of AlphaZero](https://github.com/pytorch/ELF).
> >
> > [11] Yalniz et al. 2019 [Billion-scale semi-supervised learning for image classification](https://arxiv.org/pdf/1905.00546v1.pdf).

---

> > > ### Comment · Reviewer_KruY · 2022-08-07
> > > **Thanks for the response. But some of questions are not answered in the response.**
> > >
> > > Thanks for the response. However, I am still quite curious about the questions asked in the review as below:
> > > - Do the authors plan to release the model and/or the dataset used to train the model? If so, when?
> > > - Do the authors plan to release an API like what OpenAI does to allow people to play with your model?
> > > - Can the authors discuss the differences between CM3 and Flamingo, especially the novel contributions compared to CM3 since CM3 is released months earlier and should be considered as a prior work?

---

> > > > ### Author Response · Authors · 2022-08-08
> > > > **Additional responses**
> > > >
> > > > Please find the answers to the remaining questions below:
> > > >
> > > > > Do the authors plan to release the model and/or the dataset used to train the model? If so, when?
> > > >
> > > > The dataset is proprietary and we won't be able to release it, or the models trained on it. We refer the reviewer to our previous answer on validity (trust), reproducibility and open sourcing, and the point of publishing work like this.
> > > >
> > > > > Do the authors plan to release an API like what OpenAI does to allow people to play with your model?
> > > >
> > > > We don't have such plans at the moment.
> > > >
> > > > > Can the authors discuss the differences between CM3 and Flamingo, especially the novel contributions compared to CM3 since CM3 is released months earlier and should be considered as a prior work?
> > > >
> > > > First, we would like to note that developing and training large models like Flamingo takes a significant amount of time and computational resources, and hence a few months' difference between the release is not significant. At the time of the CM3 release on arxiv, most of our models had either already been trained, or were in the process of training. In addition and to the best of our knowledge CM3 has not been published at a peer-reviewed venue yet. For these reasons, we still believe that CM3 should be considered as concurrent work. Differences between CM3 and Flamingo are listed below:
> > > >
> > > > - __1/8)__: _No visual tokenization bottleneck using VQ-VAE_: CM3 uses this discretization in order to model the joint probability of text and images while Flamingo only models the conditional probability of text given an interleaved sequence of images, videos and text. As a result, Flamingo cannot generate images while CM3 can. However, as shown in visual examples from the CM3 paper, this discretization stage is quite harmful as it removes all of the high-frequency details present in images. In fact, it is impossible to read the small text from an image decoded from its VQ-VAE tokens. CM3's author acknowledged this bottleneck in the paper: "most failure cases of our proposed zero-shot captioning are due to the loss of texture from representing images through discrete tokens (e.g., the text of the train station is blurred, as is the text on the bus).". Instead, Flamingo processes the raw, non-discretised visual inputs, which allows the model to perform well on small image details. For example, our model can read small text given the strong results on the TextVQA benchmark.
> > > >
> > > > - __2/8)__: _A more efficient cross-attention mechanism_: As CM3 treats visual input as 'text' tokens, it uses all-to-all self-attention between all text tokens and visual input. In Flamingo, visual information is added through our proposed Perceiver Resampler and our novel gated cross-attention mechanism, where we explicitly biased our attention to the last preceding image. In practice, this allows our model to scale to 32 video clips of 32 frames each (1024 images in total) in a sequence at test time while being only trained with a maximum of 5 images in a sequence. We believe our approach to be more efficient than CM3 in that respect. In fact, CM3 tokenizes each image with 256 tokens. This means that for 1024 images, a self-attention between, at least, 262144 tokens is required. This is not scalable for standard language models.
> > > >
> > > > - __3/8)__: Our work focuses on leveraging frozen large autoregressive language models (i.e. such as GPT-3, PaLM, Gopher) . This was key in our work for strong in-context few-shot learning, out of the box generalisation on tasks such as visual dialogue, visual question answering, and more importantly for efficiency of training as measured in the ablation study.
> > > >
> > > > - __4/8)__: CM3: No mention of in-context few-shot results, which is the main strength and focus of our work.
> > > >
> > > > - __5/8)__: CM3 does not present results on videos. Flamingo can seamlessly process videos. We have presented strong state-of-the-art video few-shot results on a large variety of video tasks.
> > > >
> > > > - __6/8)__: CM3 trains on post-processed HTML content while Flamingo is trained on plain text webpages. The latter may be a better match for the pretraining data used by the LMs that Flamingo builds upon.
> > > >
> > > > - __7/8)__: _Benchmarks_: Our work focuses on quantitatively evaluating our model on a wide range of vision-language tasks, as we considered 16 different tasks involving vision and language, each one an externally defined and often widely-studied benchmark with a common set of metrics. The tasks include captioning, classification, fine-grained / outside-knowledge visual question answering, visual dialogue or optical character recognition. In contrast, CM3 does not focus on visual understanding tasks, only reporting zero-shot captioning results on their CM3 dataset, a non-standard captioning benchmark.

---

> > > > > ### Author Response · Authors · 2022-08-08
> > > > > **Additional responses (continuation)**
> > > > >
> > > > > * __8/8)__: For Flamingo, we have emphasised the importance of training on a mixture of data containing paired image/video and captions as well as arbitrary interleaved sequence of image and text (i.e. web pages in the case of CM3 and Flamingo). As shown in the ablation study, training on web pages only is not sufficient to get good performance on a wide range of tasks. On the other hand, CM3 focuses on training on webpages only.

---

### Review · Ethics_Reviewer_LHE2 · 2022-08-15

**Recommendation:**

The main ethical issue can be (partially) mitigated if the authors do include additional experiments on a public dataset in the main paper for the camera ready.

More broadly, I think it is important to acknowledge that the reviewers' assessment of the merit of the work may biased due to the fact that the work was highly publicized by the authors' organization before/during the review period. In this context, I think it would be useful to have an explicit NeurIPS policy regarding the use of social media and public press releases, similarly as the policy re. Arxiv preprints, before/during the review period.

**Ethical Issues:**

Yes

**Ethics Review:**

The main ethical issue brought up by several reviewers relates to reproducibility. More specifically, the dataset used to train the model as well as the trained model are proprietary. As a result, researchers will be unable to verify/reproduce the specific results presented in the paper.

One additional point that I think it is important to bring up is that this work has been highly publicized by the authors' organization (as well as the authors) before/during the review period. As a consequence, it is very likely that all the reviewers are aware of who are the authors of Flamingo and this may bias their overall assessment both negatively (reviewer KruY) and positively (reviewers ZdRp, agGw, T9XK).

---

### Meta-Review · Area_Chair_f9T4 · 2022-08-27

**Recommendation:** Accept
**Confidence:** Certain

**Metareview:**

This paper proposed Flamingo, a visual-language pretrained model, which is built based on existing powerful pretrained pure language models and pure image models.  By fixing parameters of the existing langauge model and visual model, the proposed model is further pretrained with additional perceiver and gated cross-attention components, on  a mixture of vision and language datasets.  The model can take as input an sequence of interleaved text and image/vidios, and generate text output.  The model demonstrated its performances on a range of open-ended and close-ended visual-language tasks, in zero-shot or few-shot settings.  Reviewer KruY proposed stronge concerns on the reproducibility of the work because of not releasing the source codes and datasets and the lack of some dataset details.  Nevertheless, other reviewers all agree to accept the paper because of the contribution of the paper to the community while being aware of the reproducibility problem.  I think the paper is good enough and represent a new sota in a range of tasks in this area, and is acceptable.

**Award:**

No

---

### Decision · Program_Chairs · 2022-09-14

Accept